# Adversary Aware Optimization for Robust Defense

**Daniel Wesego**
Department of Computer Science
University of Illinois Chicago
Chicago, IL 60607
dweseg2@uic.edu

**Pedram Rooshenas**
Department of Computer Science
University of Illinois Chicago
Chicago, IL 60607
pedram@uic.edu

## Abstract

Deep neural networks remain highly susceptible to adversarial attacks, where small, subtle perturbations to input images may induce misclassification. We propose a novel optimization-based purification framework that directly removes these perturbations by maximizing a Bayesian-inspired objective combining a pretrained diffusion prior with a likelihood term tailored to the adversarial perturbation space. Our method iteratively refines a given input through gradient-based updates of a combined score-based loss to guide the purification process. Unlike existing optimization-based defenses that treat adversarial noise as generic corruption, our approach explicitly integrates the adversarial landscape into the objective. Experiments performed on CIFAR-10 and CIFAR-100 demonstrate strong robust accuracy against a range of common adversarial attacks. Our work offers a principled test-time defense grounded in probabilistic inference using score-based generative models. Our code can be found at https://github.com/rooshenasgroup/aaopt.

## 1 Introduction

Neural networks have surpassed human accuracy on many vision benchmarks, yet they remain alarmingly vulnerable to adversarial examples – worst-case perturbations that change model predictions with high confidence [Szegedy et al., 2014, Goodfellow et al., 2015, Carlini and Wagner, 2017b]. Research increasingly suggests that this fragility arises from multiple interacting geometric factors as deep models partition the input space into a mosaic of piecewise linear (or gently nonlinear) regions with large local Lipschitz constants, which means that even tiny steps can radically reorder the logits [Hein and Andriushchenko, 2017, Fawzi et al., 2018, Yang et al., 2020].

The generation of an adversarial example can be expressed as a problem of finding a minimally perturbed version of an input that changes the classification output. Given a clean image $\mathbf{x}$ with true label $y$, and a model $f$, an adversary seeks a perturbation $\boldsymbol{\gamma}$ bounded by a norm constraint $\|\boldsymbol{\gamma}\|_p \leq \epsilon$ that maximizes the classification loss:

$$\max_{\|\boldsymbol{\gamma}\|_p \leq \epsilon} \mathcal{L}\big(f(\mathbf{x} + \boldsymbol{\gamma}), y\big). \tag{1}$$

Powerful gradient-based attack methods are commonly used to find such perturbations. For instance, projected gradient descent (PGD) [Madry et al., 2018a] iteratively crafts an adversarial example by taking steps in the direction of the sign of the gradient: $\mathbf{x}^{(t+1)} = \Pi_{\|\boldsymbol{\gamma}\|_p \leq \epsilon}\big(\mathbf{x}^{(t)} + \alpha \operatorname{sign}(\nabla_{\mathbf{x}^{(t)}} \mathcal{L}(f(\mathbf{x}^{(t)}), y))\big)$. Another notable approach, the Carlini–Wagner (CW) attack [Carlini and Wagner, 2017a], solves a margin-based optimization problem to find the minimum perturbation $\boldsymbol{\gamma}$ required to induce misclassification, i.e., $f(\mathbf{x} + \boldsymbol{\gamma}) \neq y$. These methods underscore how model gradients can be exploited to identify subtle yet effective input alterations.

By ascending the input gradient of the loss, which often has a significant component orthogonal to the data manifold, gradient based attacks pierce through the fragile margins between classes. While

gradients are not strictly orthogonal, even small $\ell_\infty$ perturbations can cross decision boundaries in high-dimensional pixel spaces by targeting the most sensitive non-robust features. The structure of ReLU networks further exacerbates this vulnerability: within each activation pattern the model behaves affinely, allowing first-order optimization to be highly effective, while rapid switching between activation regions ensures that vulnerable directions are densely interleaved with benign ones [Qin et al., 2019].

Adversarial training (AT) defends a classifier by augmenting each mini-batch with worst-case perturbations and solving a min–max objective [Madry et al., 2018b]. Empirically, this paradigm remains the de facto gold standard for $\ell_\infty$ and $\ell_2$ threat models: models trained with AT are the hardest to break on standard robustness benchmarks. Subsequent variants such as TRADES [Zhang et al., 2019], MART [Wang et al., 2020], and AWP [Wu et al., 2020] explicitly balance clean accuracy and robust loss or perturb the weights to enlarge margins, partially alleviating the well-known accuracy–robustness trade-off.

Despite these gains, AT still faces significant practical hurdles. Generating multi-step adversarial examples for every mini-batch inflates training cost by $2$–$30\times$, even with accelerations like Free [Shafahi et al., 2019] and Fast AT [Wong et al., 2020]. When access is limited to a fixed pretrained network, common in industrial pipelines or third-party deployments, full adversarial retraining may be infeasible. While parameter-efficient robust adapters and adversarial fine-tuning offer partial solutions, they still incur non-trivial compute. Furthermore, robustness learned under one norm and radius does not necessarily transfer to other threat models or distribution shifts, though recent work on multi-norm training and data augmentation has made progress in this direction [Gowal et al., 2021].

A complementary line of work completely avoids retraining by purifying the input at test time. In this paradigm, the adversarial input $\hat{\mathbf{x}} = \mathbf{x} + \gamma$ is treated as a corrupted observation, and the goal is to recover a reconstruction $\tilde{\mathbf{x}}$ that both lies near $\hat{\mathbf{x}}$ and has high likelihood under a learned data prior. Early approaches such as PixelDefend [Song et al., 2018] and Defense-GAN [Samangouei et al., 2018] frame this as an optimization problem or projection onto the manifold of a generative model trained on clean data. A more general perspective is offered by the framework of Regularization by Denoising (RED) [Romano et al., 2017], which shows that any powerful denoiser implicitly defines a regularizer corresponding to an implicit data prior. Under this view, purification reduces to solving the following optimization problem:

$$\min_{\mathbf{x}} -\log p(\hat{\mathbf{x}} \mid \mathbf{x}) + \lambda \mathbf{x}^T \big[ \mathbf{x} - D(\mathbf{x}) \big], \tag{2}$$

where $\mathcal{D}(\mathbf{x})$ is the denoising function.

Modern work replaces heuristic denoisers with score-based generative models. A diffusion model is trained to predict the score $\nabla_{\mathbf{x}} \log p_t(\mathbf{x})$ under Gaussian noise at multiple scales; at test time, a stochastic sampler is run that iteratively nudges $\hat{\mathbf{x}}$ toward high-density regions [Zhang et al., 2023, Nie et al., 2022b]. Because the score network is trained solely on Gaussian-corrupted data and never sees adversarial perturbations, it implicitly treats the perturbation as generic Gaussian noise – an assumption that can fail when adversarial attacks exploit directions lying in low-probability yet classifier-sensitive subspaces.

Adversarial perturbations, in general, differ from random corruptions in that they are intentional, geometry-aware, and often exploit highly structured, non-robust directions in the input space. Crucially, these perturbations are not merely local, and they exhibit global regularities. Moosavi-Dezfooli et al. [2017] shows that a single, fixed perturbation vector can fool most inputs, implying that decision boundaries are consistently oriented across different regions of the input space. Taken together, these local (gradient-based) and global (shared-structure) properties suggest that adversarial perturbations are not arbitrary noise, but follow *statistically learnable patterns*.

This insight motivates the development of *learned perturbation models*, which aim to approximate the distribution of adversarial directions, improving purification strategies that go beyond the generic denoising distribution of adversarial directions. Our method draws inspiration from test-time optimization strategies [Zhang et al., 2023, Mardani et al., 2024, Cohen et al., 2021] that recover inputs by minimizing a tailored loss function. Rather than assuming a simple Gaussian model to approximate $p(\hat{\mathbf{x}} \mid \mathbf{x})$, we explicitly capture the true adversarial perturbation distribution by training a score-based diffusion model. This model learns the gradient of the log-likelihood under realistic attack patterns,

which we then combine with the pretrained diffusion prior's score to direct the optimization. Because training a conditional score model for the likelihood term, similar to the prior, is going to be expensive, we propose a lightweight score network to learn only the attack perturbations. This principled approach retains the strong generative power of the prior diffusion model and explicitly accounts for the adversarial structure using the perturbation model. Because it is based on optimization, it avoids the full reverse sampling process, offering a more efficient and robust defense mechanism.

Our contributions can be summarized as follows:

- We introduce a novel adversarial defense framework that employs score-based adversarial optimization with diffusion models, designed to maximize the posterior probability $p(\mathbf{x} \mid \hat{\mathbf{x}})$ of clean images given inputs that may be clean or adversarially perturbed.

- Our proposed score-based diffusion optimization formulation integrates knowledge of the adversarial landscape into the purification process, making the defense more robust to attacks. The likelihood term $p(\hat{\mathbf{x}} \mid \mathbf{x})$ is learned using an efficient, trainable perturbation model implemented as a score-based diffusion network without retraining the classifier.

- We validate our approach on CIFAR-10, CIFAR-100, and other datasets against strong adversarial attacks, achieving superior robust accuracy compared to state-of-the-art diffusion-based defenses. Our approach removes adversarial distortions while faithfully retaining the underlying image content, demonstrating its potential for practical robust systems.

## 2 Method

In this section, we describe our score-based optimization framework for adversarial purification. We begin with preliminary sections on diffusion models for adversarial purification. We then continue to present our approach, starting with the motivation and then formulating purification as a maximum a posteriori (MAP) estimation problem, deriving a tractable objective using pretrained diffusion and perturbation score networks, and then presenting an efficient iterative algorithm for test-time refinement.

### 2.1 Diffusion Models for Purification

Diffusion models [Sohl-Dickstein et al., 2015, Ho et al., 2020] (also known as score-based generative models Song and Ermon [2019], Song et al. [2021]) are a class of deep generative models that learn to synthesize data by reversing a gradual noising process. During the forward diffusion process, a data sample $\mathbf{x}_0 \sim q_{\text{data}}(\mathbf{x})$ is progressively corrupted by adding Gaussian noise according to a schedule $\{\beta_t\}_{t=1}^{T}$. After $T$ steps, $\mathbf{x}_T$ is nearly an isotropic Gaussian regardless of $\mathbf{x}_0$. A diffusion model learns the reverse denoising dynamics: starting from pure noise $\mathbf{x}_T \sim \mathcal{N}(0, I)$ to produce samples that follow the data distribution. Training of diffusion models is typically done by denoising score matching Song and Ermon [2019] or an equivalent variational bound Ho et al. [2020]. In practice, a neural network $s_\theta(\mathbf{x}_t, t)$ is trained to predict either the added noise $\epsilon_t$ or the score (gradient of the log density) at each timestep using squared error loss. Sampling then proceeds by reversing this process, either via discrete timesteps [Ho et al., 2020] or continuous SDE/ODE formulations [Song et al., 2021].

Building on this generative capability, diffusion models have become a cornerstone of adversarial purification. In this setting, a potentially perturbed input $\hat{\mathbf{x}} = \mathbf{x} + \boldsymbol{\gamma}$ is first injected with a small amount of Gaussian noise by forward diffusing it to get $\mathbf{x}_t$ and then denoised through $t$ reverse diffusion steps to produce a purified output $\tilde{\mathbf{x}} = \Phi(\hat{\mathbf{x}})$. The classifier $f$ is applied to $\tilde{\mathbf{x}}$ in hopes of recovering the original prediction, $f(\tilde{\mathbf{x}}) = f(\mathbf{x})$. While diffusion purification methods are effective at mitigating adversarial perturbations, they are not without drawbacks. The stochastic and non-differentiable nature of the reverse process can obscure gradients, yet remains vulnerable to proper adaptive attacks such as PGD-EOT [Lee and Kim, 2023].

### 2.2 Motivation

Diffusion purification methods mainly rely on an unspervised approach where a pretrained diffusion prior can be applied to a broad range of classifiers without retraining. While this classifier-agnostic

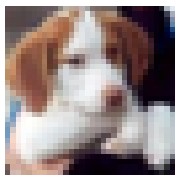
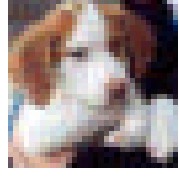
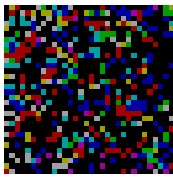

(a) Clean            (b) Adversarial            (c) Perturbation

Figure 1: Samples from the clean, adversarial, and the difference perturbation between them. Image is normalized and only showing values above a threshold value of $0.03$

approach broadens applicability, it leaves unexploited the rich structure of adversarial attacks themselves. In this work, we ask: can we augment diffusion purification with an attack-aware component that directly models the adversarial landscape, improving robustness, yet remains applicable across different classifiers? Our proposed Adversary Aware Optimization (AAOpt) answers this question by integrating a perturbation model into the diffusion framework, delivering substantial gains in defense performance without sacrificing the plug-and-play convenience of unsupervised purification.

### 2.3 Proposed Approach

Our goal is to recover the clean data given an adversarial perturbation, which is equivalent to finding the maximum a posteriori (MAP) estimation for the following problem:

$$\arg \max_{\mathbf{x}} \log p(\mathbf{x}|\hat{\mathbf{x}}) \propto \log p(\hat{\mathbf{x}}|\mathbf{x}) + \log p(\mathbf{x}). \tag{3}$$

The prior $p(\mathbf{x})$ is supplied by a fixed, pretrained diffusion model whose score function directs samples toward the natural-image manifold. Rather than ignoring the likelihood term $p(\hat{\mathbf{x}}|\mathbf{x})$ or assuming a Gaussian distribution, we explicitly train it using a *perturbation model* that estimates the residual $\mathbf{x} - \hat{\mathbf{x}}$ between clean and adversarial inputs. We model both the clean data prior $p(\mathbf{x})$ and the perturbation likelihood $p(\hat{\mathbf{x}} \mid \mathbf{x})$ with score–based diffusion models. Although we never evaluate these densities in closed form, we can query their noise-prediction networks, $\epsilon_\theta(\mathbf{x}_t, t)$, which implicitly parameterize the corresponding score functions $\boldsymbol{s}_\theta(\mathbf{x}_t, t) \approx \nabla_{\mathbf{x}_t} \log p_t(\mathbf{x}_t) \approx (-1/\sigma_t)\epsilon_\theta(\mathbf{x}_t, t)$ obtained at diffusion time $t$. For a single noise level $\sigma_t$ and corrupt $\mathbf{x}$ as $\mathbf{x}_t = \mathbf{x} + \sigma_t \epsilon,\ \epsilon \sim \mathcal{N}(\mathbf{0}, \mathbf{I})$, we can show that (see Appendix 6.1 for the derivation):

$$\log p(\mathbf{x}) \ \propto \ -\mathbf{x}^\top \big[\epsilon_\theta(\mathbf{x}_t, t) - \epsilon\big]. \tag{4}$$

Therefore, maximizing the right–hand side increases the likelihood of the data, even though $p(\mathbf{x})$ itself is never evaluated.

The surrogate in eq. 4 is reminiscent of $\log p(\mathbf{x}) \propto \mathbf{x}^T \big[D(\mathbf{x}) - \mathbf{x}\big]$, which has been used in RED [Romano et al., 2017], Mardani et al. [2024] and Zhang et al. [2023]. Mardani et al. [2024] derive similar equation for their prior from the diffusion loss $\lambda_t ||\epsilon_\theta(\mathbf{x_t}, \mathbf{t}) - \epsilon)||^2$ by using a linear loss based on the gradient under some conditions.

We use a similar approach to obtain the surrogate for $\log p(\hat{\mathbf{x}} \mid \mathbf{x})$. Let $\boldsymbol{\gamma} = \mathbf{x} - \hat{\mathbf{x}}$ be the adversarial perturbation. We simply define $p(\hat{\mathbf{x}} \mid \mathbf{x}) := p(\boldsymbol{\gamma})$.

We train the diffusion model $\boldsymbol{d}_\phi$ for the adversarial perturbations following Song et al. [2021]. Similar to eq. 4, for $\epsilon' \sim \mathcal{N}(\mathbf{0}, \mathbf{I})$ we have:

$$\log p(\boldsymbol{\gamma}) \propto -(\mathbf{x} - \hat{\mathbf{x}})^T \big[\boldsymbol{d}_\phi(\boldsymbol{\gamma}_t, t) - \epsilon'\big] \tag{5}$$

Finally, we can approximate the MAP solution in eq. 3 as:

$$\arg \max_{\mathbf{x}} \log p(\mathbf{x} \mid \hat{\mathbf{x}}) \approx \arg \max_{\mathbf{x}} -\mathbf{x}^T \big[(\epsilon_\theta(\mathbf{x}_t, t) - \epsilon) + \lambda(\boldsymbol{d}_\phi(\boldsymbol{\gamma}_t, t) - \epsilon')\big], \tag{6}$$

where $\lambda$ balances the log data prior vs log probability of the perturbation.

Algorithm 1 shows how we perform test-time adversarial purification by iteratively refining the given image $\hat{\mathbf{x}}$ toward its MAP estimate $\mathbf{x}^*$. To avoid propagating the gradients through the parameters of

---

**Algorithm 1** Adversarial Purification Optimization Algorithm

---

**Require:** $\hat{\mathbf{x}}$; pretrained diffusion net $\boldsymbol{\epsilon}_\theta$; perturbation diffusion net $\boldsymbol{d}_\phi$; total steps $M$; time range $[t_{\min}, t_{\max}]$; schedule $\{\alpha_t\}_{t=1}^T$; weight $\lambda$

1: **for** $i = 1$ **to** $M$ **do**
2:      Sample $t \sim \text{Uniform}[t_{\min}, t_{\max}]$, $\boldsymbol{\epsilon}, \boldsymbol{\epsilon}' \sim \mathcal{N}(\mathbf{0}, \mathbf{I})$                ▷ sample time and noise
3:      $\mathbf{x}_t \leftarrow \sqrt{\alpha_t}\mathbf{x}^{(i-1)} + \sqrt{1-\alpha_t}\,\boldsymbol{\epsilon}$
4:      $\ell_{\text{prior}} \leftarrow \left[\text{sg}(\boldsymbol{\epsilon}_\theta(\mathbf{x}_t, t)) - \boldsymbol{\epsilon}\right]^T \mathbf{x}^{(i-1)}$                        ▷ prior loss
5:      $\boldsymbol{\gamma} \leftarrow \mathbf{x}^{(i-1)} - \hat{\mathbf{x}}$                                      ▷ compute perturbation
6:      $\boldsymbol{\gamma}_t \leftarrow \sqrt{\alpha_t}\boldsymbol{\gamma} + \sqrt{1-\alpha_t}\,\boldsymbol{\epsilon}'$
7:      $\ell_{\text{pert}} \leftarrow \left[\text{sg}(\boldsymbol{d}_\phi(\boldsymbol{\gamma}_t, t)) - \boldsymbol{\epsilon}'\right]^T \mathbf{x}^{(i-1)}$                    ▷ perturbation loss
8:      $\ell \leftarrow \ell_{\text{prior}} + \lambda\,\ell_{\text{pert}}$                                     ▷ total loss
9:      $\mathbf{x}^{(i)} \leftarrow \text{OptimizerStep}(\mathbf{x}^{(i-1)}, \ell)$                      ▷ MAP update
10: **end for**
     **return** $\mathbf{x}^{(M)}$

---

| (a) Sample 1 | (b) Sample 2 | (c) Sample 3 |

Figure 2: Different random samples generated from the perturbation model. Image is normalized and only showing values above a threshold value of $0.03$

the score-based diffusion models during the optimization, we use stop-gradient (sg) operations when evaluating the score-based diffusion models, which will make the optimization efficient.

To train the perturbation model $\boldsymbol{d}_\phi$, at each iteration, we sample a minibatch of clean images $\mathbf{x}_{\text{clean}}^i$ and pair each with a randomly chosen sample $\mathbf{x}_k^i$ from the combined set of clean and adversarial examples length $K$. We combine clean samples in the combined set because $\hat{\mathbf{x}}$ can be either adversarial or clean in real test settings. Adversarial examples are generated only once using a combination of attacks on the classifier ($l_\infty, l_2$). We form the residual $\boldsymbol{\gamma}^i = \mathbf{x}_{\text{clean}}^i - \mathbf{x}_k^i$ and then diffuse it to timestep $t$ by adding noise $\boldsymbol{\epsilon}$ according to the diffusion schedule $\alpha_t$ and train the network as a normal diffusion model. The details of the training can be found in the Appendix section 6.4. Figure 1 shows clean and adversarial samples from the CIFAR-10 dataset with the adversarial perturbation plotted next to them. Figure 2 shows samples from the perturbation model that capture the patterns of the perturbation of the attack landscape. For the prior score network, any diffusion model trained to generate clean data can be used.

We refer to our full test-time defense as Adversary Aware Optimization (**AAOpt**). In AAOpt, the perturbation score diffusion network $\boldsymbol{d}_\phi$ explicitly captures the adversarial likelihood $p(\hat{\mathbf{x}} \mid \mathbf{x})$ by learning to predict the residual at multiple noise levels, while the pretrained score diffusion $\boldsymbol{\epsilon}_\theta$ provides a strong prior $p(\mathbf{x})$ over natural images. At inference, AAOpt iteratively refines the input via a small number of gradient-based updates: each update minimizes a combined loss that blends the diffusion prior and perturbation terms. This compact, MAP-driven loop delivers strong robustness against fully adaptive PGD-EOT and BPDA-EOT attacks, without modifying the underlying classifier or resorting to expensive reverse diffusion sampling.

## 3 Related Work

A variety of defense strategies have been proposed to tackle adversarial attacks. Early purification approaches used simple image transformations, such as bit depth reduction or compression, to remove high-frequency noise, but these filter-based methods offered limited robustness and were easily bypassed by adaptive attacks [Xu et al., 2018]. Generative-model defenses then leveraged GANs or

VAEs to project inputs onto a learned manifold of clean images; DefenseGAN [Samangouei et al., 2018] optimizes a latent code to reconstruct a purified image, yet it introduces significant inference latency and can suffer from mode collapse.

Energy-based models (EBMs) were also applied to adversarial purification by Hill et al. [2021], using long-run Langevin dynamics to project adversarial inputs onto the natural data manifold and averaging over multiple stochastic samples via Expectation-Over-Transformation (EOT). However, contrastive divergence training and MCMC sampling make EBMs slow and unstable. Yoon et al. [2021] replaced EBMs with score-based networks trained by denoising score matching [Song and Ermon, 2019], and proposed a deterministic sampling technique with only a few Langevin iterations. They also improved robustness by injecting noise into inputs before purification.

The advent of diffusion priors ushered in more powerful purification techniques. DiffPure [Nie et al., 2022a] demonstrated that adding a small amount of noise to an adversarial example followed by a fixed number of reverse diffusion steps can effectively remove perturbations, achieving robust accuracy. However, the reverse chain often comprising of more than 50 timesteps entails high computational cost and may also drift the sample's semantics away from the original content depending on the selected timestep $t$. ScoreOpt [Zhang et al., 2023] improved efficiency by performing test-time optimization using the diffusion model's score function at random noise levels to guide a small number of gradient updates rather than executing the entire sampling chain. Chen et al. [2023] integrates classification directly into the diffusion process to jointly denoise and classify. They first optimize the input to remove adversarial perturbations and then use a diffusion classifier to perform the classification. Lin et al. [2024] proposed a guidance method on top of diffusion purification based on a distance metric between the features of adversarial examples and forward diffused $\mathbf{x_t}$ to improve robustness.

Concurrently, works on diffusion-based inverse problems have introduced various formulations for solving both linear and non-linear measurement tasks. Song et al. [2021] showed that diffusion models can be applied to inverse problems via a controllable generation process, sampling from $p(\mathbf{x}_t \mid \mathbf{y})$ using Bayes' rule and factoring the score as $\nabla_{\mathbf{x}_t} \log p_t(\mathbf{x}_t) + \nabla_{\mathbf{x}_t} \log p_t(\mathbf{y} \mid \mathbf{x}_t)$. Kawar et al. [2022] proposed Denoising Diffusion Restoration Models (DDRM), which provides an efficient unsupervised method for solving inverse tasks by utilizing pretrained denoising diffusion models and guidance from the observed input $\mathbf{y}$. The guidance is performed by considering the SVD of the degradation matrix and noise levels, which are then incorporated into the sampling process. Mardani et al. [2024] introduced RED-Diff, another method of optimization to solve inverse problems on images using diffusion prior models. They propose a variational lower bound that consists of a measurement matching loss and a score-based regularization term with pretrained diffusion models.

# 4 Experiments

We evaluate our score-based adversarial optimization method mainly on CIFAR-10 and CIFAR-100 datasets [Krizhevsky et al., 2009], benchmarking against a suite of strong and adaptive white-box attacks and established defense baselines. All experimental results are reported on the test splits, and we report both the clean (standard) accuracy and the robust accuracy when the model is under attack. We compare our approach to different Adversarial training (AT) methods [Pang et al., 2022, Gowal et al., 2021] and different diffusion purification methods [Nie et al., 2022a, Zhang et al., 2023, Chen et al., 2023]. We reuse some results from already implemented works [Lee and Kim, 2023, Zhang et al., 2023], while we reimplement Score-Opt-O and Score-Opt-N methods with the same pretrained diffusion model and classifier checkpoints as our proposed method.

We employ a pretrained diffusion model using the EDM preconditioning scheme of Karras et al. [2022], trained to minimize $\mathbb{E}_{\mathbf{x} \sim p_{\text{data}}} \mathbb{E}_{\mathbf{n} \sim \mathcal{N}(0, \sigma^2 I)} \left\| D(\mathbf{x} + \mathbf{n}, \sigma) - \mathbf{x} \right\|^2$, where $D(\cdot, \sigma)$ predicts a denoised estimate at each noise level. We extract the predicted noise from the model, where $\boldsymbol{\epsilon}_\theta = (\mathbf{x}_t - D(\mathbf{x}_t))/\sigma_t$. For all methods, we use a WideResNet-28-10 or a WideResNet-70-16 classifier [Zagoruyko and Komodakis, 2016] trained on clean data, except where baselines specify adversarial training. To train our perturbation score model, we first generate adversarial examples for the classifiers using a gradient attack with the cross-entropy loss from the training set once. Unless otherwise mentioned, we use $l_\infty$ and $l_2$ norms with a combination of $\epsilon = (2, 4, 8)$ for $l_\infty$ and $\epsilon = (0.5, 1)$ for $l_2$ norm attacks. Those will be combined with the clean data to train the perturbation model using the DDPM diffusion setup with 1000 discrete steps [Ho et al., 2020].

We evaluate robustness under multiple adaptive attacks. We first show our results on the PGD-EOT (Projected Gradient Descent with Expectation Over Transformation) with $l_\infty$ budget $\epsilon = 8/255$ and $l_2$ budget $\epsilon = 0.5$ with 20 steps and 20 EOT samples per step [Athalye et al., 2018]. We then use the BPDA-EOT (Backward Pass Differentiable Approximation) attack setup with 50 BPDA iteration steps and 15 EOT samples per step. These attacks are crafted against the full purification (optimization) with the classification pipeline to measure end-to-end robustness of the models. We follow a similar experimental setup with Lee and Kim [2023], Zhang et al. [2023]. Remaining hyperparameter details and additional experiments can be found in the Appendix.

Table 1: Standard and robust accuracy against PGD+EOT on CIFAR-10. Left: $l_\infty(\epsilon = 8/255)$; Right: $l_2(\epsilon = 0.5)$. Compared with adversarial training (AT) and adversarial purification (AP) methods.

| Type | Method | Standard | Robust | Type | Method | Standard | Robust |
|---|---|---|---|---|---|---|---|
| **WideResNet-28-10** | | | | **WideResNet-28-10** | | | |
| | Default | 94.78 | 0.0 | | Default | 94.78 | 0.0 |
| AT | Pang et al. [2022] | 88.62 | 64.95 | AT | Sehwag et al. [2021] | 90.93 | 83.75 |
| | Gowal et al. [2020] | 88.54 | 65.10 | | Rebuffi et al. [2021] | 91.79 | 85.05 |
| | Gowal et al. [2021] | 87.51 | 66.01 | | Augustin et al. [2020] | 93.96 | 86.14 |
| AP | Yoon et al. [2021] | 85.66 | 37.27 | AP | Yoon et al. [2021] | 85.66 | 74.26 |
| | Nie et al. [2022a] | 90.07 | 51.25 | | Nie et al. [2022a] | 91.41 | 82.11 |
| | Score-Opt-O[2023] | 91.21 | 64.96 | | Score-Opt-O[2023] | 91.21 | 79.09 |
| | Score-Opt-N[2023] | 94.43 | 65.62 | | Score-Opt-N[2023] | 94.43 | 84.86 |
| | Lin et al. [2024] | 90.42 | 64.06 | | Lin et al. [2024] | 90.42 | 85.55 |
| | AAOpt (ours) | $91.91_{\pm 0.8}$ | $\mathbf{86.75}_{\pm 0.7}$ | | AAOpt (ours) | $91.91_{\pm 0.8}$ | $\mathbf{87.61}_{\pm 0.6}$ |
| **WideResNet-70-16** | | | | **WideResNet-70-16** | | | |
| | Default | 95.19 | 0.0 | | Default | 95.19 | 0.0 |
| AT | Gowal et al. [2021] | 88.75 | 69.03 | AT | Rebuffi et al. [2021] | 92.41 | 86.24 |
| | Wang et al. [2023] | 92.97 | 72.46 | | Wang et al. [2023] | 96.09 | 86.72 |
| AP | Yoon et al. [2021] | 86.76 | 41.02 | AP | Yoon et al. [2021] | 86.76 | 75.90 |
| | Nie et al. [2022a] | 90.43 | 57.03 | | Nie et al. [2022a] | 92.15 | 84.80 |
| | Chen et al. [2023] | 87.89 | 71.68 | | Chen et al. [2023] | 87.89 | 75.00 |
| | Score-Opt-O[2023] | 92.61 | 68.86 | | Score-Opt-O[2023] | 92.61 | 79.55 |
| | Score-Opt-N[2023] | 95.11 | 70.2 | | Score-Opt-N[2023] | 95.11 | 85.21 |
| | AAOpt (ours) | $91.59_{\pm 0.7}$ | $\mathbf{86.24}_{\pm 0.8}$ | | AAOpt (ours) | $91.59_{\pm 0.7}$ | $\mathbf{87.49}_{\pm 0.7}$ |

**Table 1** reports the results of clean and robust accuracy on the CIFAR10 dataset when performing the PGD-EOT attack. The PGD-EOT attack is performed under the budgets of $l_\infty$ budget $\epsilon = 8/255$ and $l_2$ budget $\epsilon = 0.5$. We use 20 steps of the optimization algorithm shown in Algorithm 1. When generating adversarial attacks, we approximate the gradients of our model by performing a single step of the exact optimization method, which will ensure the attack has full knowledge and access to all components of our defense. We compare against leading adversarial training (AT) and diffusion-based purification (AP) baselines using two backbone classifiers (WRN-28-10 and WRN-70-16). Across both architectures and attack norms, our adversary-aware optimization consistently delivers the highest robust accuracy, often outperforming prior defenses by many points, while maintaining clean accuracy on par with the best competing methods. These results demonstrate that our proposed optimization method offers state-of-the-art resilience to adaptive adversaries without sacrificing clean performance.

**Table 2** highlights the effectiveness of different gradient-approximation strategies in the optimization loop, all evaluated on CIFAR-10 ($\ell_\infty$, $\epsilon = 8/255$) with a WRN-28-10 classifier. Using a single step of our exact forward-optimization update yields the highest robust accuracy of $86.75\%$, showing the strength of our adversary-aware refinement. Even when omitting the stop-gradient (i.e., allowing gradients to backpropagate through the score networks) when generating adversarial examples, we still achieve $84.08\%$ robust accuracy which substantially outperforms the other baselines shown in Table 1. In addition to this, we compare our approach with ADBM [Li et al., 2025], which finetunes a diffusion model with adversarial data, under different settings as detailed in Section 6.3 of the

Appendix. Finally, we use a single step of denoising through the prior diffusion to approximate the gradients similar to Zhang et al. [2023], which also attains a robust accuracy of 75.11% that exceeds the best diffusion-purification and adversarial-training methods. These results underscore that our core optimization framework is both powerful and resilient, delivering leading adversarial robustness results.

Table 2: Robust accuracy on CIFAR-10 ($\ell_\infty$, $\epsilon = 8/255$) for different gradient approximation methods using the WRN-28-10 classifier.

| Gradient Approximation Method | Robust Accuracy (CIFAR-10, $\epsilon = 8/255$) |
|---|---|
| One step of exact forward optimization | $86.75_{\pm 0.7}$ |
| One step of forward optimization with no stop-grad | $84.08_{\pm 0.3}$ |
| One step of denoising | $75.11_{\pm 1.2}$ |

**Table 3a** compares clean and robust accuracy under a strong BPDA-EOT attack on $\ell_\infty$ with $\epsilon = 8/255$ across adversarial training (AT), diffusion-based purification (AP), and our proposed adversary-aware score optimization method. Adversarial Training methods plateau around 64% robust accuracy ( Gowal et al. [2020] achieves 64.1%) while diffusion purifiers substantially improve robustness with Score-Opt-N pushing the result to 90.1%. Our approach further advances the result by achieving a robust accuracy of 91.4%, while maintaining a clean accuracy of 92.1%. These results demonstrate that our proposed method withstands different kinds of adaptive attacks but also outperforms prior works on adversarial training and purification defenses. **Table 3b** shows the respective results for CIFAR-100 dataset. Our method is also robust on this data, achieving a robust accuracy of 66.56%, while maintaining a standard accuracy of 69.98%, which outperforms other diffusion purification methods under the same attack conditions on robustness with comparable standard accuracy.

Table 3: BPDA+EOT attack on CIFAR-10 and CIFAR-100 ($\ell_\infty$, $\epsilon = 8/255$) threat models (all architectures WRN-28-10).

(a) CIFAR-10 (WRN-28-10)

| Type | Method | Std. | Rob. |
|---|---|---|---|
| | Default | 94.78 | 0.0 |
| AT | Carmon et al. [2019] | 89.67 | 63.10 |
| | Gowal et al. [2020] | 89.48 | 64.08 |
| AP | Wang et al. [2022] | 93.50 | 79.83 |
| | Yoon et al. [2021] | 86.14 | 70.01 |
| | Nie et al. [2022a] | 89.02 | 81.40 |
| | Score-Opt-O[2023] | 90.23 | 81.36 |
| | Score-Opt-N[2023] | 93.94 | 90.07 |
| | AAOpt (ours) | $92.10_{\pm 0.7}$ | $\mathbf{91.36_{\pm 0.8}}$ |

(b) CIFAR-100 (WRN-28-10)

| Type | Method | Std. | Rob. |
|---|---|---|---|
| | Default | 81.55 | 0.0 |
| AP | Yoon et al. [2021] | 60.66 | 39.72 |
| | Hill et al. [2021] | 51.66 | 26.10 |
| | Score-Opt-O[2023] | 70.53 | 66.11 |
| | Score-Opt-N[2023] | 74.18 | 60.21 |
| | AAOpt (ours) | 69.98 | **66.56** |

**Table 4** compares a baseline purification pipeline, omitting our perturbation-score correction and only using the prior pretrained diffusion score-regularization method against our full AAOpt method on CIFAR-10 under both PGD-EOT and BPDA-EOT attacks with a WRN-70-16 and WRN-28-10 classifiers, respectively. Incorporating our adversary-aware perturbation score model boosts clean accuracy by at least 1% and yields substantial gains in robust accuracy under both PGD-EOT and BPDA-EOT attacks. Figure 3 shows how the standard and robust accuracies change as $\lambda$ is increased from 0 to 1 on WRN-28-10 BPDA-EOT attack. The accuracies start to increase initially and plateau around $\lambda = 0.6$ and then decrease.

**Generalization** Table 5 evaluates the transferability of our perturbation model across different classifier architectures. We compare performance on WRN-70-16 when using a perturbation model trained on its own adversarial examples versus one trained on WRN-28-10. The minimal drop in accuracy demonstrates that our learned attack distribution generalizes effectively to a classifier it was not originally trained on. Table 6 also shows the robustness of our model against different unseen corruptions [Hendrycks and Dietterich, 2019]. We evaluate it against CIFAR10-C data on a

Table 4: Baseline (without perturbation-score model correction) similar to score-regularization of Mardani et al. [2024] vs. Full Method (AAOpt) comparison on CIFAR-10 ($\ell_\infty$, $\epsilon = 8/255$) under PGD-EOT using WRN-70-16 and BPDA-EOT using WRN-28-10.

| Method | PGD-EOT | | BPDA-EOT | |
|---|---|---|---|---|
| | Standard | Robust | Standard | Robust |
| Baseline | $89.99_{\pm0.7}$ | $82.88_{\pm0.9}$ | $91.05_{\pm0.6}$ | $81.95_{\pm0.6}$ |
| AAOpt (ours) | $91.59_{\pm0.7}$ | $86.24_{\pm0.8}$ | $92.10_{\pm0.7}$ | $91.36_{\pm0.8}$ |

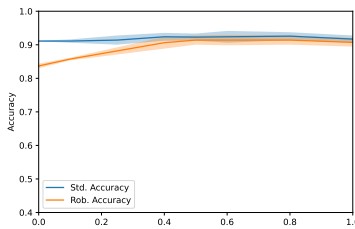

Figure 3: Accuracy as $\lambda$ is increased from 0 to 1 for AAOpt on BPDA-EOT on CIFAR-10 ($\ell_\infty$, $\epsilon = 8/255$)

Table 5: Transferability of perturbation models: accuracy on WRN-70-16 using perturbation models trained on WRN-70-16 and WRN-28-10 adversarial data under PGD-EOT attacks ($\ell_\infty$, $\epsilon = 8/255$ and $\ell_2$, $\epsilon = 0.5$).

| Perturbation Model | PGD-EOT ($\ell_\infty$, $\epsilon = 8/255$) | | PGD-EOT ($l_2$, $\epsilon = 0.5$) | |
|---|---|---|---|---|
| | Standard | Robust | Standard | Robust |
| Perturbation Model of WRN-70-16 | $91.59_{\pm0.7}$ | $86.24_{\pm0.8}$ | $91.59_{\pm0.7}$ | $87.49_{\pm0.7}$ |
| Perturbation Model of WRN-28-10 | $91.51_{\pm1.1}$ | $85.90_{\pm0.9}$ | $91.51_{\pm1.1}$ | $87.20_{\pm0.8}$ |

Table 6: Robustness against common corruptions on CIFAR10-C on WRN-70-16

| Method | gaussian | elastic transform | jpeg compression | snow | brightness |
|---|---|---|---|---|---|
| Default | 45.6 | 82.9 | 75.8 | 83.5 | 93.8 |
| Score-Opt-O | 79.8 | 78.5 | 86.1 | 82.67 | 89.6 |
| AAOpt | $85.3_{\pm0.2}$ | $84.81_{\pm0.3}$ | $89.5_{\pm0.1}$ | $85.3_{\pm0.4}$ | $90.5_{\pm0.1}$ |

combination of five different severity levels of transformations. These results confirm that AAOpt's perturbation-aware optimization yields strong, classifier-agnostic defenses across both adversarial and natural distribution shifts.

Beyond the transfer experiments in Table 5, which apply a WRN-28-10 trained perturbation model to a WRN-70-16 classifier, we further assess AAOpt with WRN-28-10 trained perturbation model on a CLIP zero-shot model applied to CIFAR-10. We compare against the adversarially trained Robust-CLIP FARE[2] and FARE[4] models [Schlarmann et al., 2024], which were trained with $\ell_\infty$ budgets of $\epsilon = 2/255$ and $\epsilon = 4/255$ respectively as shown in Table 7. Despite never seeing CLIP during the perturbation model training, AAOpt, using only the WRN-28-10 perturbation model, achieves superior robust accuracy on this entirely different architecture, underscoring the wide applicability and classifier-agnostic nature of our adversary-aware optimization. In addition, it exhibits consistent performance across different attack budgets, unlike adversarial training models, which drastically lower their performance when attacked with a budget different from the one on which the model was trained.

Finally, we present experiments on norm-transferability. We train a perturbation model using a single norm and evaluate it on a different norm to assess generalization across attack types. As shown in Table 8, the performance remains consistent, indicating that the proposed approach exhibits strong transferability between $\ell_\infty$ and $\ell_2$ perturbation spaces. Specifically, when trained on $\ell_2(\epsilon = 0.5)$ and evaluated with an $\ell_\infty(\epsilon = 8/255)$ PGD-EOT attack, the model achieves a robust accuracy of 86.22% which is close to the results reported on Table 1 where the perturbation model is trained on the combination of the two norm attacks. Similar performance is also reported when trained on $\ell_\infty(\epsilon = 8/255)$ and evaluated under $\ell_2(\epsilon = 0.5)$ perturbations. These results demonstrate that the perturbation model generalizes well across different norm constraints, confirming the robustness and adaptability of our method.

Table 7: Transfer-PGD attack on CIFAR-10 ($\ell_\infty$, $\epsilon = 4/255$ and $\epsilon = 8/255$) on the CLIP classifier using perturbation model of WRN-28-10 classifier.

| Type | Method | Standard | Robust ($\epsilon = 4/255$) | Robust ($\epsilon = 8/255$) |
|------|--------|----------|-----------------------------|-----------------------------|
|      | CLIP   | 95.2     | 0.0                         | 0.0                         |
| AT   | FARE$^2$ | 89.5   | 25.9                        | 2.22                        |
|      | FARE$^4$ | 77.7   | 34.6                        | 9.01                        |
| AP   | Score-Opt-O[2023] | 89.21 | 88.74              | 88.70                       |
|      | Score-Opt-N[2023] | 90.42 | 89.87              | **90.70**                   |
|      | AAOpt (ours) | $91.52_{\pm0.9}$ | $\mathbf{91.63}_{\pm1.3}$ | $90.30_{\pm1.1}$ |

Table 8: Norm transferability: accuracy of WRN-28-10 using perturbation models trained on a different norm under PGD-EOT attacks ($\ell_\infty$, $\epsilon = 8/255$ and $\ell_2$, $\epsilon = 0.5$).

| Perturbation Model Trained On | Evaluated On | Rob. Acc. |
|-------------------------------|--------------|-----------|
| $\ell_\infty$, $\epsilon = 8/255$ | $\ell_2$, $\epsilon = 0.5$) | 87.07 |
| $\ell_2$, $\epsilon = 0.5$)   | $\ell_\infty$, $\epsilon = 8/255$ | 86.22 |

We report additional results and implementation details in Appendix, including theoretical derivations, extended robustness evaluations (AutoAttack and ADBM comparisons), perturbation analyses, classifier transfer-attack results, and experiments on TinyImageNet. The appendix also provides full training and dataset specifications, including hyperparameters and compute resources, and concludes with a discussion of the limitations of this work.

## 5  Conclusion and Discussion

We have introduced **Adversary Aware Optimization (AAOpt)**, a novel framework for test-time defense that explicitly incorporates both a diffusion prior and a learned perturbation model. By training a perturbation model to capture the adversarial landscape and combining it with a pretrained diffusion score model, AAOpt performs a small number of gradient-based refinement steps to recover purified inputs. Extensive experiments on CIFAR-10 and CIFAR-100 under strong, adaptive PGD-EOT and BPDA-EOT attacks demonstrate that AAOpt consistently outperforms state-of-the-art adversarial training and purification baselines, achieving robust accuracy improvements while maintaining high clean accuracy. The limitations of our work are that AAOpt requires generating adversarial examples once for training the perturbation model and incurs additional test-time cost due to a few gradient-based refinement iterations. However, these overheads are modest compared to reverse-diffusion sampling or repeated classifier retraining, and the benefit of robust accuracy across multiple norms and attacks, coupled with minimal impact on clean performance, far outweighs the costs. Our framework is also highly modular: the perturbation model and diffusion prior can be improved independently. Consequently, AAOpt represents a practical path toward robust, test-time defenses in real-world settings.

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

# 6 Appendix

This appendix provides supplementary material to the main text. We first derive the surrogate equation in section 6.1. In Section 6.2.1 we report AutoAttack results; Section 6.2.2 compares our results with ADBM [Li et al., 2025]; Section 6.2.3 analyzes the perturbation patterns of adversarial examples; Section 6.2.4 presents transfer-attack experiments; and Section 6.2.5 shows results on TinyImageNet. Next, we will discuss the datasets used in Section 6.3. We then detail our experimental setup, including hyperparameters, optimization, training, and compute resources in Section 6.4, and conclude with an extended discussion of limitations in Section 6.5.

## 6.1 Derivation of Surrogate Loss

Assuming $\mathbf{x}_t = \mathbf{x} + \sigma_t \boldsymbol{\epsilon}$ and $\boldsymbol{\epsilon} \sim \mathcal{N}(0, I)$, and the true marginal distribution at time $t$ is $p_t(\mathbf{x}_t)$, the reverse conditional distribution is modeled as

$$p_\theta(\mathbf{x} \mid \mathbf{x}_t) = \mathcal{N}\big(\mathbf{x}; \boldsymbol{\mu}_\theta(\mathbf{x}_t, t), \sigma_t^2 \boldsymbol{I}\big),$$

with

$$\boldsymbol{\mu}_\theta(\mathbf{x}_t, t) = \mathbf{x}_t - \sigma_t \boldsymbol{\epsilon}_\theta(\mathbf{x}_t, t).$$

1. From Baye's identity for posterior log-density:

$$\log p(\mathbf{x}) = \log p_t(\mathbf{x}_t) + \log p_\theta(\mathbf{x} \mid \mathbf{x}_t) - \log q(\mathbf{x}_t \mid \mathbf{x}),$$

where

$$q(\mathbf{x}_t \mid \mathbf{x}) = \mathcal{N}(\mathbf{x}_t; \mathbf{x}, \sigma_t^2 \boldsymbol{I}).$$

Note that

$$-\log q(\mathbf{x}_t \mid \mathbf{x}) = \frac{1}{2\sigma_t^2} \|\mathbf{x}_t - \mathbf{x}\|^2 + \text{const} = \frac{1}{2} \|\boldsymbol{\epsilon}\|^2 + \text{const}.$$

Therefore,

$$\log p(\mathbf{x}) = \log p_t(\mathbf{x}_t) + \log p_\theta(\mathbf{x} \mid \mathbf{x}_t) + \frac{1}{2} \|\boldsymbol{\epsilon}\|^2 + \text{const}. \tag{7}$$

2. From Taylor's expansion of $\log p_t(\mathbf{x})$ around $\mathbf{x}_t$:

$$\log p_t(\mathbf{x}) \approx \log p_t(\mathbf{x}_t) + (\mathbf{x} - \mathbf{x}_t)^T \nabla_{\mathbf{x}_t} \log p_t(\mathbf{x}_t).$$

Rearranged as:

$$\log p_t(\mathbf{x}_t) \approx \log p_t(\mathbf{x}) - (\mathbf{x} - \mathbf{x}_t)^T \nabla_{\mathbf{x}_t} \log p_t(\mathbf{x}_t).$$

Substitute $\log p_t(\mathbf{x}_t)$ in eq. 7:

$$\log p(\mathbf{x}) \approx \log p_t(\mathbf{x}) - (\mathbf{x} - \mathbf{x}_t)^T \nabla_{\mathbf{x}_t} \log p_t(\mathbf{x}_t) + \log p_\theta(\mathbf{x} \mid \mathbf{x}_t) + \frac{1}{2} \|\boldsymbol{\epsilon}\|^2 + \text{const}.$$

which can be written as:

$$\log p(\mathbf{x}) \propto \sigma_t \boldsymbol{\epsilon}^T \nabla_{\mathbf{x}_t} \log p_t(\mathbf{x}_t) + \log p_\theta(\mathbf{x} \mid \mathbf{x}_t) + \frac{1}{2} \|\boldsymbol{\epsilon}\|^2. \tag{8}$$

3. Expressing $\log p_\theta(\mathbf{x} \mid \mathbf{x}_t)$ using diffusion model $\epsilon_\theta$:

$$\log p_\theta(\mathbf{x} \mid \mathbf{x}_t) \propto -\frac{1}{2\sigma_t^2} \|\mathbf{x} - \boldsymbol{\mu}_\theta(\mathbf{x}_t, t)\|^2 + \text{const},$$

where

$$\boldsymbol{\mu}_\theta(\mathbf{x}_t, t) = \mathbf{x}_t - \sigma_t \boldsymbol{\epsilon}_\theta(\mathbf{x}_t, t).$$

Note that

$$\mathbf{x} - \boldsymbol{\mu}_\theta = \mathbf{x} - (\mathbf{x}_t - \sigma_t \boldsymbol{\epsilon}_\theta) = (\mathbf{x} - \mathbf{x}_t) + \sigma_t \boldsymbol{\epsilon}_\theta = -\sigma_t \boldsymbol{\epsilon} + \sigma_t \boldsymbol{\epsilon}_\theta = \sigma_t (\boldsymbol{\epsilon}_\theta - \boldsymbol{\epsilon}).$$

Therefore,

$$\log p_\theta(\mathbf{x} \mid \mathbf{x}_t) \propto -\frac{1}{2}\|\boldsymbol{\epsilon}_\theta - \boldsymbol{\epsilon}\|^2 + \text{const.} \tag{9}$$

4. Substituting $\log p_\theta(\mathbf{x} \mid \mathbf{x}_t)$ back in eq. 8:

$$\log p(\mathbf{x}) \propto \sigma_t \boldsymbol{\epsilon}^T \nabla_{\mathbf{x}_t} \log p_t(\mathbf{x}_t) - \frac{1}{2}\|\boldsymbol{\epsilon}_\theta - \boldsymbol{\epsilon}\|^2 + \frac{1}{2}\|\boldsymbol{\epsilon}\|^2 + \text{const.}$$

Expanding the quadratic term,

$$-\frac{1}{2}\|\boldsymbol{\epsilon}_\theta - \boldsymbol{\epsilon}\|^2 + \frac{1}{2}\|\boldsymbol{\epsilon}\|^2 = -\frac{1}{2}\|\boldsymbol{\epsilon}_\theta\|^2 + \boldsymbol{\epsilon}^T \boldsymbol{\epsilon}_\theta.$$

Thus,

$$\log p(\mathbf{x}) \propto \sigma_t \boldsymbol{\epsilon}^T \nabla_{\mathbf{x}_t} \log p_t(\mathbf{x}_t) + \boldsymbol{\epsilon}^T \boldsymbol{\epsilon}_\theta - \frac{1}{2}\|\boldsymbol{\epsilon}_\theta\|^2 + \text{const.}$$

$$\log p(\mathbf{x}) \propto \boldsymbol{\epsilon}^T (\sigma_t \nabla_{\mathbf{x}_t} \log p_t(\mathbf{x}_t) + \boldsymbol{\epsilon}_\theta) - \frac{1}{2}\|\boldsymbol{\epsilon}_\theta\|^2$$

$$\log p(\mathbf{x}) \propto \frac{(\mathbf{x}_t - \mathbf{x})^T}{\sigma_t}(\sigma_t \nabla_{\mathbf{x}_t} \log p_t(\mathbf{x}_t) + \boldsymbol{\epsilon}_\theta) - \frac{1}{2}\|\boldsymbol{\epsilon}_\theta\|^2$$

$$\log p(\mathbf{x}) \propto \frac{\mathbf{x}_t^T}{\sigma_t}(\sigma_t \nabla_{\mathbf{x}_t} \log p_t(\mathbf{x}_t) + \boldsymbol{\epsilon}_\theta) - \frac{\mathbf{x}^T}{\sigma_t}(\sigma_t \nabla_{\mathbf{x}_t} \log p_t(\mathbf{x}_t) + \boldsymbol{\epsilon}_\theta) - \frac{1}{2}\|\boldsymbol{\epsilon}_\theta\|^2$$

Dropping the non $\mathbf{x}$ terms:

$$\log p(\mathbf{x}) \propto -\frac{(\mathbf{x})^T}{\sigma_t}(\sigma_t \nabla_{\mathbf{x}_t} \log p_t(\mathbf{x}_t) + \boldsymbol{\epsilon}_\theta)$$

$$\log p(\mathbf{x}) \propto -\mathbf{x}^T(\sigma_t \nabla_{\mathbf{x}_t} \log p_t(\mathbf{x}_t) + \boldsymbol{\epsilon}_\theta)$$

Finally, we can replace true score by the conditional Gaussian score in expectation to make it tractable Zhou et al. [2024]:

$$\nabla_{\mathbf{x}_t} \log p_t(\mathbf{x}_t) \approx \nabla_{\mathbf{x}_t} \log q(\mathbf{x}_t \mid \mathbf{x}) = -\frac{1}{\sigma_t}\boldsymbol{\epsilon}.$$

$$\log p(\mathbf{x}) \propto -\mathbf{x}^T(\boldsymbol{\epsilon}_\theta(\mathbf{x}_t, t) - \boldsymbol{\epsilon})$$

Which is the same as eq 4. Similarly, the loss will be:

$$-\log p(\mathbf{x}) \propto \mathbf{x}^T(\boldsymbol{\epsilon}_\theta(\mathbf{x}_t, t) - \boldsymbol{\epsilon})$$

which is the same loss used by Mardani et al. [2024] for their prior, but they formulated it in a different way.

## 6.2 Additional Results

### 6.2.1 AutoAttack

In addition to PGD-EOT and BPDA-EOT attacks presented in the main experiment, we assess the robustness of our model using AutoAttack [Croce and Hein, 2020], an ensemble of complementary, parameter-free attacks that provides a strong, automated benchmark for robustness. We conduct these evaluations on a WRN-28-10 classifier, generating adversarial examples via an exact one-step gradient through our optimization loop. Table 9 presents AutoAttack robustness alongside PGD-EOT robust accuracy for direct comparison; baseline results are taken from Zhang et al. [2023], Lee and Kim [2023], Lin et al. [2024]. Although running the full multi-step optimization for AutoAttack yields very similar performance, repeating those experiments across multiple seeds and attack configurations is prohibitively time-consuming, so we report the one-step exact-gradient results here. This evaluation confirms that AAOpt maintains high robust accuracy even under this rigorous, adaptive attack suite, demonstrating its superior performance against diverse baselines and attacks.

Table 9: AutoAttack and PGD-EOT on CIFAR-10 ($\ell_\infty$, $\epsilon = 8/255$) using the WRN-28-10 classifier.

| Type | Method | Standard | PGD | AutoAttack |
|---|---|---|---|---|
| WideResNet-28-10 | | | | |
| | Default | 94.78 | 0.0 | 0.0 |
| AT | Pang et al. [2022] | 88.62 | 64.95 | 61.04 |
| | Gowal et al. [2020] | 88.54 | 65.10 | 62.76 |
| | Gowal et al. [2021] | 87.51 | 66.01 | 63.38 |
| AP | Yoon et al. [2021] | 85.66 | 37.27 | 59.53 |
| | Nie et al. [2022a] | 90.07 | 51.25 | 63.6 |
| | Lee and Kim [2023] | 90.16 | 55.82 | 70.47 |
| | Lin et al. [2024] | 90.42 | 64.06 | 78.12 |
| | Ours (AAOpt) | $91.91_{\pm 0.8}$ | $\mathbf{86.75}_{\pm 0.7}$ | $\mathbf{88.28}_{\pm 0.5}$ |

### 6.2.2 Comparison with ADBM

In general, our optimization has a stop gradient on the diffusion models during optimization to remove adversarial perturbations. PGD-based gradient-based attacks are also generated this way (remember we need a second-order derivative for generating attacks), where gradients don't pass through the diffusion model, similar to our optimization mechanism. While this is ok, a stronger attack can be generated by backpropagating through the score-based diffusion models. In the main text, we added Table 2, which shows the different performance of our model under different gradient approximations, even with no stop-gradient. In this section, we want to compare our work with ADBM [Li et al., 2025], which modifies the diffusion objective itself to guide adversarial examples toward the clean distribution. They also propose a full gradient attack where they show the performance of different diffusion purification methods perform very poorly. Unlike diffusion purification works discussed in [Li et al., 2025], AAOpt performs very well in this scenario also. The results are summarized in Table 10 and reveal that AAOpt significantly outperforms ADBM in robust accuracy. The experiments were conducted under a similar setup to ADBM for a fair comparison with PGD-EOT, with 20 attack iterations, setup similar to the one found in their appendix, and BPDA-EOT with 200 attack iterations.

The PGD results for AAOpt are with gradients passing through both diffusion models to generate adversarial data, approximated with one optimization step. Even with full gradient passing through all the steps, similar to the ADBM full grad setup for the PGD, we get a robust accuracy of $\approx 82\%$, further confirming its effectiveness even under stronger attack variants. But due to the computation time and the resources it takes to get the full gradients with no stop-grad through all the optimization steps, we approximate it with one step, which provides close results but is more efficient to conduct.

Table 10: Comparison with ADBM on CIFAR-10 ($\ell_\infty$, $\epsilon = 8/255$) using the WRN-28-10 classifier.

| Method | BPDA-EOT | | PGD-EOT | |
|---|---|---|---|---|
| | Standard | Robust | Standard | Robust |
| ADBM | 91.93 | 70.51 | 92.50 | 42.20 |
| Ours (AAOpt) | $92.10_{\pm 0.7}$ | $\mathbf{90.89}_{\pm 1.2}$ | $91.91_{\pm 0.8}$ | $\mathbf{84.08}_{\pm 0.3}$ |

### 6.2.3 Attack Patterns

In this section, we analyze the structure of adversarial perturbations and their relationship to the original images. Figure 4 displays a clean image, its adversarially perturbed counterpart, and the difference between them. Although these difference patterns may appear noisy at first glance, they exhibit systematic correlations with the underlying content of the clean image. Capturing these consistent perturbation patterns is the key motivation for our perturbation-score network. As illustrated by the samples in Figure 2, our model learns to represent and predict these adversarial residuals.

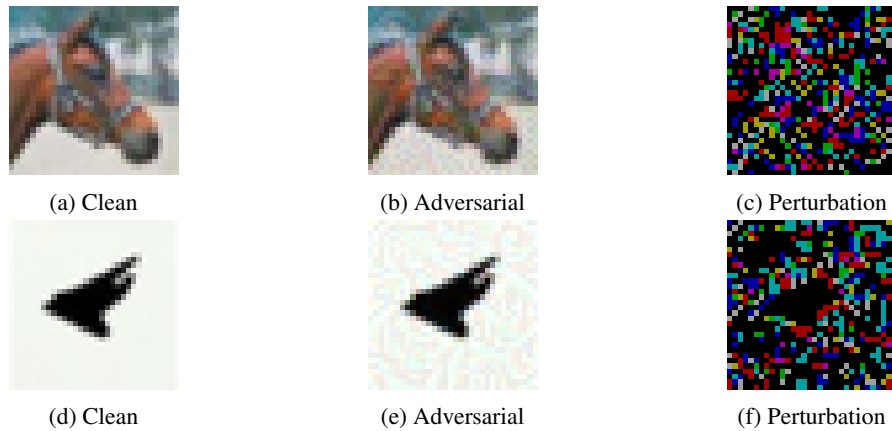

|          |          |          |
|:--------:|:--------:|:--------:|
| (a) Clean | (b) Adversarial | (c) Perturbation |
| (d) Clean | (e) Adversarial | (f) Perturbation |

Figure 4: Clean, Adversarial, and Perturbation of a sample image from CIFAR-10, perturbation is normalized and only showing values above a threshold value of $0.03$

### 6.2.4 Transfer PGD Attack

We evaluate transfer-based robustness by generating $\ell_\infty$-bounded PGD adversarial examples on the classifier alone and then applying various purification methods—including our AAOpt framework—to these fixed attacks. Table 11 reports clean and robust accuracy for each purifier. Our AAOpt method delivers robustness on par with the strongest diffusion purifiers, closely following Score-Opt-N, which achieves the highest robust accuracy in this setting. These results confirm that AAOpt's adversary-aware modeling substantially improves resilience to transfer attacks without sacrificing clean accuracy.

Table 11: Transfer-PGD attack on CIFAR-10 ($\ell_\infty$, $\epsilon = 8/255$) using the WRN-28-10 classifier.

| Method | Standard | Robust |
|--------|:--------:|:------:|
| Default | 94.78 | 0.0 |
| Yoon et al. [2021] | 93.09 | 85.45 |
| Score-Opt-O[2023] | 89.79 | 87.45 |
| Score-Opt-N[2023] | 92.65 | **92.76** |
| Ours (AAOpt) | $92.14_{\pm 0.6}$ | $91.01_{\pm 0.5}$ |

### 6.2.5 TinyImageNet Results

To demonstrate that AAOpt scales to more complex, higher-dimensional data, we evaluate on TinyImageNet – a 200-class subset of ImageNet with images resized to 64×64 [Deng et al., 2009]. We fine-tune a ResNet-50 classifier [He et al., 2016] on this dataset and attack it using BPDA-EOT with 50 iterations and 15 EOT samples. Table 12 compares our method against Score-Opt-O baseline. Even in this challenging setting, AAOpt achieves the highest robust accuracy, confirming its effectiveness on larger, more diverse benchmarks. The result can be improved further with more powerful classifiers and better prior models but we leave that for future work.

Table 12: BPDA-EOT attack on TinyImageNet ($\ell_\infty$, $\epsilon = 8/255$ using ResNet50 classifier.

| Method | Standard | Robust |
|--------|:--------:|:------:|
| ResNet50 | 76.90 | 0.0 |
| Score-Opt-O[2023] | 40.89 | 21.2 |
| Ours (AAOpt) | $56.10_{\pm 1.9}$ | $\mathbf{53.75}_{\pm 1.5}$ |

## 6.3 Datasets

Our evaluations leverage several widely used benchmark datasets. We use CIFAR-10 and CIFAR-100 [Krizhevsky et al., 2009], each comprising 50 000 training and 10 000 test images at 32×32 resolution; CIFAR-10 spans 10 classes, while CIFAR-100 covers 100 classes. These datasets are publicly available from `https://www.cs.toronto.edu/~kriz/cifar.html`, though no explicit license is specified by the authors.

To assess robustness under common corruptions, we employ CIFAR-10-C [Hendrycks and Dietterich, 2019], which augments the original CIFAR-10 test set with different types of corruptions at five severity levels. CIFAR-10-C is released under a CC BY 4.0 International license (DOI: `https://doi.org/10.5281/zenodo.2535967`).

Finally, we include TinyImageNet200, a 200-class subset of ImageNet containing 100 000 training and 10 000 validation images resized to 64×64. While the Stanford CS231N release (available at `http://cs231n.stanford.edu/tiny-imagenet-200.zip`) does not list its own license, it inherits the ImageNet terms of use, which permit non-commercial research and educational use only [Deng et al., 2009].

## 6.4 Experiment details

In this subsection, we cover the experimental details of training and inference for our proposed model. We first list the hyperparameters in the model, then we go over optimization, training, and compute details. We have also attached the code used to generate the outputs, which can be found in the attached supplementary material for more details. We run the main experiments in the paper 5 times and report the mean and standard deviation. We use 512 samples for each trial because of the heavy computation requirements to generate adversarial examples.

### 6.4.1 Hyperparameters

This section details the hyperparameters for our AAOpt framework. For each attack setting, we specify the Adam learning rate, the number of optimization iterations, the weight $\lambda$ balancing the diffusion prior and perturbation model losses, and the number of forward diffusion timesteps. Table 13 summarizes all selected values.

Table 13: Details of hyperparameters used for our proposed model AAOpt

| Attack Type | Perturbation Budget | LR | Iterations | $\lambda$ | Timesteps |
|---|---|---|---|---|---|
| CIFAR-10 | | | | | |
| PGD+EOT | $l_\infty(\epsilon = 8/255)$ | 0.1 | 20 | 0.25 | Uniform (0.15,0.35) |
| PGD+EOT | $l_2(\epsilon = 0.5)$ | 0.1 | 20 | 0.25 | Uniform (0.15,0.35) |
| AutoAttack | $l_\infty(\epsilon = 8/255)$ | 0.1 | 20 | 0.25 | Uniform (0.15,0.35) |
| BPDA+EOT | $l_\infty(\epsilon = 8/255)$ | 0.1 | 5 | 0.5 | [0.5, ..., 0] |
| Transfer-PGD | $l_\infty(\epsilon = 8/255)$ | 0.02 | 20 | 0.6 | [0.2, ..., 0] |
| Transfer-PGD-CLIP | $l_\infty(\epsilon = 8/255)$ | 0.01 | 20 | 0.2 | [0.1, ..., 0] |
| CIFAR-10-Corrputions | | | | | |
| Corruptions | – | 0.1 | 5 | 0.1 | [0.25, ..., 0] |
| CIFAR-100 | | | | | |
| BPDA+EOT | $l_\infty(\epsilon = 8/255)$ | 0.1 | 3 | 0.4 | [0.15, ..., 0] |

### 6.4.2 Optimization

The core purification loop is detailed in Algorithm 1, where at each iteration we update the current estimate $\mathbf{x}$ by taking an Adam step on the combined loss. To make the entire procedure end-to-end differentiable, so that an adaptive attacker can compute gradients through our optimization update, we implement a custom differentiable Adam optimizer in PyTorch rather than relying on the built-in version, which uses in-place operations and is not differentiable. In addition, when generating

adversarial examples via a single exact optimization step, we set the create_graph flag to True, which yields the correct gradients of the adversary's loss with respect to $\mathbf{x}$ through the optimization. Without this, attacks like PGD-EOT would treat the optimization updates as constants, potentially generating a weaker form of attack.

### 6.4.3 Training

We used a pretrained EDM diffusion for the prior model [Karras et al., 2022] for both the CIFAR-10 and CIFAR-100 datasets. For training the perturbation model, we use a UNET architecture with three input and output channels. We train the model according to Ho et al. [2020] with unweighted diffusion loss. The residual difference between the clean and adversarial images is diffused to a random timestep $t^i$ by adding noise $\boldsymbol{\epsilon}^i$ according to the diffusion schedule $\{\alpha_t\}$. The network $\boldsymbol{d}_\phi$ takes the noisy perturbation of the residual $\boldsymbol{\gamma}_t^i$ and predicts the injected noise $\hat{\boldsymbol{\epsilon}}^i$. We then minimize the denoising objective $\|\hat{\boldsymbol{\epsilon}}^i - \boldsymbol{\epsilon}^i\|^2$ and train the network as a normal diffusion DDPM model over the perturbation to capture the score-based diffusion of the residual distribution at varying noise levels. Note that the residual $\boldsymbol{\gamma}^i$ can also be defined in the reverse direction as $\boldsymbol{\gamma}^i = \mathbf{x}_{\text{adv}}^i - \mathbf{x}_{\text{clean}}^i$ with the same setup as the negative sign in the difference will cancel out with the perturbation loss term. The details of the training are shown in Algorithm 2.

To train our perturbation score model, we generate adversarial examples once for the classifiers. We use a gradient attack with the cross-entropy loss from the training set. We use $l_\infty$ and $l_2$ attack norms with a combination of $\epsilon = (2, 4, 8)$ for $l_\infty$ and $\epsilon = (0.5, 1)$ for $l_2$ attack budgets. We combine these with the clean data, and one adversarial example is randomly sampled uniformly to form the pair for training the perturbation model.

For the classifiers, we used a WideResNet architecture classifiers with WRN-28-10 and WRN-70-16 types [Zagoruyko and Komodakis, 2016]. For the CIFAR-10 dataset, the WRN-28-10 architecture yields an accuracy of 94.78, while the WRN-70-16 architecture yields an accuracy of 95.19. The robust accuracy of these default classifiers is approximately 0% for both $l_\infty$ and $l_2$ attack budgets of $\epsilon = 8/255$ and 0.5 respectively.

---

**Algorithm 2** Perturbation Model Training

---

**Require:** Clean dataset $\{\mathbf{x}_{\text{clean}}\}$, combined dataset of adversarial and clean dataset $\{\mathbf{x}^k\}_{k=1}^K$ (where each $\mathbf{x}^k$ is either clean or adversarial), score network $\boldsymbol{d}_\phi$, diffusion timesteps $T$, schedule $\{\alpha_t\}_{t=1}^T$
 1: **for** each training iteration **do**
 2:     Sample $\{\mathbf{x}_{\text{clean}}^i\}_{i=1}^B$ from clean data and $\mathbf{x}_{\text{adv}}^i$ from $\{\mathbf{x}_{\text{adv}}^k\}_{k=1}^K$
 3:     Find perturbation $\boldsymbol{\gamma}^i = \mathbf{x}_{\text{clean}}^i - \mathbf{x}_{\text{adv}}^i$
 4:     Sample timestep $t^i \sim \text{Uniform}\{1, \ldots, T\}$
 5:     Sample noise $\boldsymbol{\epsilon}^i \sim \mathcal{N}(\mathbf{0}, \boldsymbol{I})$
 6:     Compute noisy diff: $\boldsymbol{\gamma}_t^i = \sqrt{\bar{\alpha}_t}\, \boldsymbol{\gamma}^i + \sqrt{1 - \bar{\alpha}_t}\, \boldsymbol{\epsilon}^i$
 7:     Predict: $\hat{\boldsymbol{\epsilon}}^i = \boldsymbol{d}_\phi(\boldsymbol{\gamma}_t^i, t^i)$
 8:     Compute loss: $\ell^i = \|\hat{\boldsymbol{\epsilon}}^i - \boldsymbol{\epsilon}^i\|^2$
 9:     Update Parameters: $\boldsymbol{d}_\phi$
10: **end for**

---

### 6.4.4 Compute

We use NVIDIA A100 GPU to train the models and perform the inference. Training the perturbation UNET model takes approximately 6GB of GPU memory for a batch size of 256. We trained the perturbation model for 3000 epochs, and one epoch approximately takes 18 seconds. We use the Adam optimizer with a constant learning rate of 2e-5. More details and the exact training code can be found in the attached supplementary material.

Table 14 shows how much time the optimization algorithm takes compared to Score-Opt-* baselines tested on our environment on an A100 GPU. The iteration steps for the optimizations on this table are 5 steps on the CIFAR-10-C dataset, which is similar in characteristics to the CIFAR-10 dataset. As seen in the table, our model is slightly faster compared to the baselines. Note that the time shown is for the optimization to purify $\mathbf{x}$, and not for generating adversarial examples.

Table 14: Optimization speed of our model compared with baselines on CIFAR-10-C

| Model | Batch Size | Time Consumed |
|---|---|---|
| Score-Opt-O | 128 | 1.16 seconds |
| Score-Opt-N | 128 | 1.26 seconds |
| Ours (AAOpt) | 128 | 1.05 seconds |

## 6.5 Limitation and Discussion

While AAOpt achieves state-of-the-art robustness, it relies on several assumptions and incurs non-trivial computational costs that may limit its performance. First, our perturbation model requires generation of adversarial examples once. Although we have shown this to be enough based on our experiments as tested in different scenarios, the effectiveness of our perturbation model may degrade if the attacker uses a substantially different threat model or if the perturbation distribution shifts too much in practice. Second, test-time optimization requires multiple gradient-based refinement steps (3–20 iterations), which increases inference latency compared to a standard forward pass models. Third, AAOpt depends on a well-trained diffusion prior; if the diffusion model is poorly matched to the target domain or exhibits artifacts under certain noise schedules, the denoising updates may drift the image away from the true manifold. Finally, while our experiments span CIFAR-10, CIFAR-100, TinyImageNet with different classifiers of WideResNet and a CLIP zero-shot classifier, further evaluation on diverse modalities, and real-time scenarios is needed to fully characterize AAOpt's generality and scalability. Future work will focus on improving the model and extending this work to multimodal generative [Wesego and Rooshenas, 2024] models.

