# OpenReview forum: "Adversary Aware Optimization for Robust Defense"
_NeurIPS.cc/2025/Conference — NeurIPS 2025 poster_

### Official Review · Reviewer_9umZ · 2025-06-22

**Clarity:** 3
**Significance:** 2
**Originality:** 2
**Rating:** 4
**Confidence:** 2

**Summary:**

This work introduces AAOpt, an adversarial purification method for test-time adversarial defense. Compared with existing purification methods, the key difference/novelty lies in that AAOpt considers an extra diffusion model that explicitly characterizes the adversarial perturbations (rather than "normal" diffusion models that assume a gaussian perturbation). Experiments on CIFAR-10 and CIFAR-100 demonstrates improved results compared with adversarial training and other purification baselines.

**Questions:**

Please see Strengths and Weaknesses.

**Ethical Concerns:**

["NO or VERY MINOR ethics concerns only"]

**Final Justification:**

My questions are addressed in the rebuttal, and I don't have any major concern left. I raise my score from 3 to 4.

**Limitations:**

yes

**Quality:**

2

**Strengths And Weaknesses:**

While my expertise is in adversarial attacks and adversarial training, I don't closely follow the line of works on diffusion-based adversarial purification, so I may not be able to accurately assess the strengths/significance and weaknesses of this work.

### Strengths
1. It definitely makes sense to consider adversary-aware diffusion models for purification-based defense.

### Weaknesses
1. The setup in Table 5 regarding transferablity is too optimistic, i.e., models from the same architecture family (WRN) is being used. I think it's important to evaluate transferability between more distinct models, as I would imagine that the proposed method (which trains the diffusion model with certain attacks generated with a certain network) would be less transferable than previous purification-based methods that use pre-trained diffusion models.

### Questions

1. The perturbation model $d_\phi$ is trained with $\ell_\infty$ and $\ell_2$ attacks. What if the test-time attacks are with different norm? What would be the consideration here to make it generalize?
2. Is it a standard to evaluate purification methods with EOT attacks? What about AutoAttack?

---

> ### Author Rebuttal · Authors · 2025-07-31
>
> Dear Reviewer,
>
> Thank you for your thoughtful review. We address your concerns below.
>
> We submitted our appendix as a zip file with the supplementary materials, following one of the two allowed methods for NeurIPS submissions. Although we understand you are not obligated to review it, some content couldn’t fit in the main paper, necessitating its inclusion there.
>
> *Comment: "The setup in Table 5 regarding transferablity is too optimistic, i.e., models from the same architecture family (WRN) is being used. I think it's important to evaluate transferability between more distinct models, as I would imagine that the proposed method (which trains the diffusion model with certain attacks generated with a certain network) would be less transferable than previous purification-based methods that use pre-trained diffusion models." and
> Q1: "The perturbation model $d_{\phi}$ is trained with $l_2$ and $l_\infty$
> attacks. What if the test-time attacks are with different norm? What would be the consideration here to make it generalize?*
>
> We have thought about that and have already addressed some of your concerns in the Appendix. In Table 9, we study the transferability of our attack using a completely different classifier architecture of a CLIP one-shot model. Our model performs better than the baselines. This experiment, in addition to Table 5, shows that using a specific perturbation model is a better way to generalize to different classifiers and attack techniques.
>
> In addition to that, to answer your Question 1, we have added experiments on norm transferability. We train a perturbation only on one type of norm and evaluate it on a different norm. The results are very close to the one trained using joint attack norms. The results are shown below:
>
> --------------------------------------------------
>  |       AAOpt Clean Acc |    91.57                  |
>
>  |    AAOpt Robust Acc  |    86.22                   |
>
> Tab1: AAOpt Performance on $l_\infty(\epsilon = \frac{8}{255})$ PGD-EOT attack using a perturbation model trained on $l_2 (\epsilon =0.5)$ using WRN-28-10
>
>  --------------------------------------------------
>  |       AAOpt Clean Acc |    91.71                   |
>
>  |    AAOpt Robust Acc  |    87.07                   |
>
>  Tab2: AAOpt Performance on $l_2 (\epsilon =0.5)$ PGD-EOT attack using a perturbation model trained on $l_\infty(\epsilon = \frac{8}{255})$ using WRN-28-10
>
>
> *2. "Is it a standard to evaluate purification methods with EOT attacks?"*
>
> Yes, it's a standard technique and very needed as diffusion-based methods vary, and averaging over multiple transformations ensures accurate results [1].
>
> *3. " What about AutoAttack? "*
>
> We have already added AutoAttack results in the Appendix because we ran out of space in the main paper. Table 7 demonstrates the results during AutoAttack, but this attack is not as strong as the PGD-EOT attack [1]. Our model performs well in this evaluation too.
>
> We hope we have answered your questions, and we kindly ask you to reconsider the score if this has addressed your concerns. We invite you to also look at the appendix, if possible.
>
> 1. Lee, M., & Kim, D. (2023). Robust evaluation of diffusion-based adversarial purification. In Proceedings of the IEEE/CVF International Conference on Computer Vision (pp. 134-144).

---

> > ### Comment · Reviewer_9umZ · 2025-08-02
> > **Follow-up question**
> >
> > Thanks for the rebuttal.
> >
> > In Tab1 and Tab2 of the rebuttal, I only see AAOpt's result. What baseline result should I compare it with? Also, beyond showing empirical results, what I really want to understand is where this generalization across perturbation norms is coming from, especially compared with other diffusion-purification-based methods that are not adversary-aware?

---

> > > ### Author Response · Authors · 2025-08-02
> > > **Reply**
> > >
> > > Thank you for the follow-up.
> > >
> > > Tab 1 of the rebuttal should be directly compared to Table 1, top left, from the main paper $l_\infty$ attacks on $\epsilon=8/255$. The robust accuracy is 86.22 for this compared to the robust accuracy of the main paper, which is 86.75, and is trained on a combination of norms. Tab 2 of the rebuttal should be compared to Table 1, top right, from the main paper. The robust accuracy of this is also a little lower, 87.07, compared to the main paper one, which is 87.61, trained on a combination of norms.
> > >
> > > Regarding your second question, the generalization is coming from learning from the patterns of the adversarial perturbations, which are provided by the perturbation model. Adversarial perturbations differ fundamentally from random corruptions: they are deliberately crafted, sensitive to the input's geometry, and often exploit structured, non-robust directions in the data space. Importantly, these perturbations are not purely local—they display global patterns. As demonstrated in [1], a single fixed perturbation can deceive a wide range of inputs, indicating that decision boundaries are similarly aligned across different parts of the input space. These observations—of both local (gradient-driven) and global (shared structure) characteristics—suggest that adversarial perturbations are not random noise, but exhibit statistically learnable structure, which is what the perturbation model tries to capture.
> > >
> > > [1] Moosavi-Dezfooli, S. M., Fawzi, A., Fawzi, O., & Frossard, P. (2017). Universal adversarial perturbations. In Proceedings of the IEEE conference on computer vision and pattern recognition (pp. 1765-1773).

---

> ### Comment · Reviewer_9umZ · 2025-08-03
> **Thank you**
>
> I realized that I didn't phrase my question clear enough in my last comment, but now by looking at the result comparison my question is already answered. I will raise my score to 4.
>
> As a final note, just as a recommendation, I think it would be interesting to explore more on the good generalization of the proposed method across perturbation norms, either in this version or in the future. Specifically, I'm quite surprised that it generalizes well (with only minor robust accuracy degradation) when training on a single norm compared with training on multiple norms. This is in contrast with adversarial training, which in my impression is very sensitive to the mismatch between train-time and inference-time perturbation norm.

---

> > ### Author Response · Authors · 2025-08-05
> > **Acknowledgement**
> >
> > Dear reviewer, we appreciate your review. Thank you for your feedback.

---

### Official Review · Reviewer_PpXk · 2025-07-03

**Clarity:** 2
**Significance:** 2
**Originality:** 1
**Rating:** 4
**Confidence:** 3

**Summary:**

This paper proposes a novel diffusion model–based purification method to defend against adversarial perturbations. The authors identify two limitations of prior work: adversarial training incurs high retraining costs, and existing purification approaches via diffusion model without retraining implicitly treat adversarial perturbations as generic Gaussian noise, resulting in insufficient robustness. To address this, the authors introduce a new score-based optimization framework. Specifically, they train a perturbation score network to estimate the distribution of adversarial directions and leverage it to integrate knowledge of the adversarial landscape into the purification process.

**Questions:**

• Compared to methods that integrate adversarial perturbation information into the purification process, what novel aspects does this approach introduce?
• How does robustness change when the attack settings or datasets used at inference differ from those in training? Does the defense generalize across such variations?
• How much larger is AAOpt’s inference cost compared to existing methods?
• Could this framework be extended to provide certified defenses?

**Ethical Concerns:**

["NO or VERY MINOR ethics concerns only"]

**Final Justification:**

Thank you very much for providing the additional experiments. My concerns have been fully addressed. I will raise my score.

**Limitations:**

The authors claim that inference-time cost is high, but it is unclear how much overhead there is. Thus, the inference cost should be compared with existing methods.

**Quality:**

2

**Strengths And Weaknesses:**

Strength:
This paper clearly identifies the weakness in existing purification methods that implicitly treat adversarial perturbations as Gaussian noise, and proposes a lightweight score-based approach to address it. Empirical results demonstrate its effectiveness across some settings, and the provided code ensures reproducibility.

Weakness:
The technical contribution of the proposed method is incremental. Previous studies have already applied score‐based diffusion models to adversarial purification [1, 2], and other works have incorporated adversarial‐perturbation information directly into the purification process [3, 4]. Furthermore, the paper does not cite [3, 4] and fails to explain how its method differs from theirs or to provide any experimental performance comparison.

Minor Comments
• The proposed method still depends on training the perturbation‐score network with adversarial examples, making it vulnerable to unseen threat models and distribution shifts. Consequently, robustness may degrade when the attack budget or dataset changes. However, such robustness under varying conditions is not evaluated in the paper.
• There is no quantitative assessment of the inference‐time cost. While the authors acknowledge this as a limitation, they do not provide experimental measurements.
• The proposed method lacks theoretical guarantees. Some prior diffusion-based purification methods provide formal robustness proofs [5].

[1] Nie, Weili, et al. "Diffusion Models for Adversarial Purification." International Conference on Machine Learning. PMLR, 2022.
[2] Zhang, Boya, Weijian Luo, and Zhihua Zhang. "Enhancing adversarial robustness via score-based optimization." Advances in Neural Information Processing Systems 36 (2023): 51810-51829.
[3] Li, Xiao, et al. "ADBM: Adversarial diffusion bridge model for reliable adversarial purification." ICLR. 2025.
[4] Bai, Mingyuan, et al. "Diffusion models demand contrastive guidance for adversarial purification to advance." Proceedings of the 41st International Conference on Machine Learning. 2024.
[5] Zhang, Jiawei, et al. "{DiffSmooth}: Certifiably robust learning via diffusion models and local smoothing." 32nd USENIX Security Symposium (USENIX Security 23). 2023.

---

> ### Author Rebuttal · Authors · 2025-07-31
>
> Dear Reviewer,
>
> Thank you for your thoughtful review. We address your concerns below.
>
> We submitted our appendix as a zip file with the supplementary materials, following one of the two allowed methods for NeurIPS submissions. Although we understand you are not obligated to review it, some content couldn’t fit in the main paper, necessitating its inclusion there. The appendix answers most of your questions, and we hope you look at it.
>
> Your questions and minor comments overlap for the most part. So, we will try to answer them together as much as possible.
>
> *1. "Compared to methods that integrate adversarial perturbation information into the purification process, what novel aspects does this approach introduce? "*
>
> *"The technical contribution of the proposed method is incremental. Previous studies have already applied score‐based diffusion models to adversarial purification, and other works have incorporated adversarial‐perturbation information directly into the purification process ..."*
>
> We believe our method introduces a novel and distinctive contribution to adversarial purification. Specifically, we are the first to propose incorporating a perturbation model within a MAP (Maximum A Posteriori) optimization framework to explicitly maximize the likelihood of clean data during purification.
>
> Our model uniquely combines:
>
>   - A prior diffusion model, trained on clean data, which captures the data manifold accurately.
>
>   - A modular perturbation model, trained to capture the common structures of adversarial perturbations, which serves as a flexible guide toward clean data.
>
> This perturbation model enhances purification performance by learning the attack distribution. It can be adjusted via a weighting parameter λ, or even removed entirely, offering flexibility depending on the scenario.
>
> In contrast, related works such as [2] (which resembles [3] in our citations) employ guidance mechanisms during reverse diffusion, and [4] modifies the diffusion objective itself to guide adversarial examples toward the clean distribution. This method typically requires retraining the diffusion model and generating adversarial examples over many iterations for training the model, unlike ours.
>
> Our approach avoids these drawbacks by first keeping the diffusion prior intact, ensuring robustness across general settings. Then, our modular and lightweight perturbation model can generalize to different attacks. Our results also outperform existing methods, as demonstrated in our empirical results. We believe this formulation provides a more flexible, efficient, and principled way to integrate adversarial information into the optimization process.
>
> *2. "How does robustness change when the attack settings or datasets used at inference differ from those in training? Does the defense generalize across such variations?"*
>
> Our approach exhibits strong robustness and generalization across varying attack settings and classifier architectures.
>
> **Classifier Transferability**: Table 5 in the main paper shows that our perturbation model trained with one classifier (e.g., WRN-28-10) continues to perform well when applied to a different classifier (WRN-70-16), despite architectural differences. While these two models are relatively similar, we went further to assess robustness more thoroughly.
>
> **Cross-Architecture Generalization**: To test generalization across significantly different classifier families, Table 9 reports results when the perturbation model trained on WRN-28-10 is applied to a CLIP zero-shot classifier—a radically different architecture. The performance remains strong, demonstrating our method’s generalization beyond similar backbones.
>
> **Cross-Norm Generalization**: We also evaluated how a perturbation model trained under one threat model generalizes to another. Specifically, a model trained on $l_\infty$-bounded attacks ($\epsilon = \frac{8}{255}$) performs well when tested on $l_2$-bounded attacks ($\epsilon = 0.5$), confirming its cross-norm robustness.
>
> -----------------------------------------------
>  |       AAOpt Clean Acc |    91.57             |
>
>  |    AAOpt Robust Acc  |    86.22            |
>
> Tab1: AAOpt Performance on $l_\infty(\epsilon = \frac{8}{255})$ PGD-EOT attack using a perturbation model trained on $l_2 (\epsilon =0.5)$
>
>  -----------------------------------------------
>  |       AAOpt Clean Acc |    91.71              |
>
>  |    AAOpt Robust Acc  |    87.07              |
>
>  Tab2: AAOpt Performance on $l_2 (\epsilon =0.5)$ PGD-EOT attack using a perturbation model trained on $l_\infty(\epsilon = \frac{8}{255})$
>
> **Robustness to Natural Corruptions**: Additionally, Table 6 in the main paper shows our defense is effective not just against adversarial examples, but also against common corruptions such as noise, blur, and weather-based distortions.
>
> Overall, these experiments confirm that our method generalizes effectively across classifier architectures, perturbation norms, and natural distribution shifts.
>
> *3. "How much larger is AAOpt’s inference cost compared to existing methods?"*
>
> This question is addressed in the ablation study (Table 12 in the appendix).
>
> We compare AAOpt to Score-Opt [1], which also purifies adversarial examples using iterative optimization.
>
> Score-Opt [1] has demonstrated that it is faster than reverse diffusion-based methods (e.g., DiffPure), since optimization requires fewer steps than full reverse sampling.
>
> Despite being iterative, our method is faster than Score-Opt for purifying adversarial examples, due to the efficiency of our optimization procedure, as shown in Table 12.
>
> *4. "Could this framework be extended to provide certified defenses? Some prior diffusion-based purification methods provide formal robustness proofs."*
>
> Our current focus is on optimizing the posterior likelihood to remove adversarial perturbations and move samples closer to the clean distribution.
>
> At this time, our method does not provide certified robustness guarantees, but we view this as a valuable direction for future work and plan to explore it.
>
> We hope the responses above help clarify your questions. Many of these points are addressed in the appendix, but we’ve expanded them here for clarity. We would be happy to respond to any follow-up questions you may have, and we kindly ask you to reconsider the score if this has addressed your concerns.
>
> [1] Zhang, B., Luo, W., & Zhang, Z. (2023). Enhancing adversarial robustness via score-based optimization. Advances in Neural Information Processing Systems, 36, 51810-51829.
>
> [2] Bai, Mingyuan, et al. "Diffusion models demand contrastive guidance for adversarial purification to advance." Proceedings of the 41st International Conference on Machine Learning. 2024.
>
> [3] Lin, G., Tao, Z., Zhang, J., Tanaka, T., & Zhao, Q. (2025). Adversarial guided diffusion models for adversarial purification. Neural Networks, 107705.
>
> [4] Li, X., Sun, W., Chen, H., Li, Q., Liu, Y., He, Y., ... & Hu, X. (2024). Adbm: Adversarial diffusion bridge model for reliable adversarial purification. ICLR 2025.

---

> ### Comment · Area_Chair_JEDc · 2025-08-05
>
> Dear Reviewer PpXk,
>
> The deadline for author-reviewer discussion period is approaching. We kindly ask you to review the authors' rebuttal. Please provide your feedback soon. Thank you.
>
> Best,
>
> AC

---

> > ### Author Response · Authors · 2025-08-05
> > **Follow up**
> >
> > Dear reviewer,
> >
> > We hope our rebuttal has answered your questions. We will wait for your feedback.

---

> > > ### Comment · Reviewer_PpXk · 2025-08-06
> > > **Still have questions regarding the comparison with prior work**
> > >
> > > Thank you very much for your review. Some of my concerns have been resolved. However, I still have questions regarding the comparison with prior work.
> > >
> > > What is your rationale for not comparing your proposed method with [3]? While [3] requires retraining the diffusion model, this is a lightweight fine-tuning. Moreover, it does not require adversarial example generation over multiple iterations. Do you have any experimental or structural evidence that your method is significantly more lightweight or robust?

---

> > > > ### Author Response · Authors · 2025-08-06
> > > > **Response to Reviewer**
> > > >
> > > > Dear Reviewer,
> > > >
> > > > Thank you for your response.
> > > >
> > > > We provide a direct comparison between ADBM and our proposed method (AAOpt) under two attack settings: PGD-EOT and BPDA-EOT. All experiments were conducted under a similar setup to ADBM for a fair comparison. The results are summarized in the following tables and show that AAOpt significantly outperforms ADBM in robust accuracy.
> > > >
> > > > Table 1: BPDA-EOT ($\ell_\infty$, $\epsilon=8/255$, 200 iterations, 20 EOT), using ADBM settings from their OpenReview discussion.
> > > >
> > > > | Metric     | AAOpt | ADBM  |
> > > > | ---------- | ----- | ----- |
> > > > | Clean Acc  | 92.10 | 91.93 |
> > > > | Robust Acc | 89.25 | 70.51 |
> > > >
> > > >
> > > > Table 2: PGD-EOT ($\ell_\infty$, $\epsilon=8/255$, 20 iterations), ADBM results taken from Appendix Table A2 from their paper.
> > > >
> > > > | Metric     | AAOpt | ADBM  |
> > > > | ---------- | ----- | ----- |
> > > > | Clean Acc  | 91.91 | 92.50 |
> > > > | Robust Acc | 84.08 | 42.20 |
> > > >
> > > >
> > > > Table 2 presents PGD with gradients passing through both diffusion models to generate adversarial data approximated with one step for AAOpt. Even with full gradient passing through all the steps, similar to the ADBM full grad setup for the PGD, we get a robust accuracy of 81.25%, further confirming its effectiveness even under stronger attack variants. Clearly, the results show that our approach demonstrates the robustness and superiority of AAOpt over ADBM. We appreciate your thoughtful feedback, and we believe these improvements meaningfully strengthen the contribution of our work.
> > > >
> > > > Please let us know if you have any further questions or would like additional details. We’d be happy to provide them.

---

> > > > > ### Author Response · Authors · 2025-08-08
> > > > > **Follow up**
> > > > >
> > > > > Dear reviewer,
> > > > >
> > > > > We hope our reply has answered your questions. Just following up to ensure our earlier response and additional results addressing your concerns were seen. We believe the clarifications and experiments provided meaningfully strengthen our work, and we’re happy to provide anything further if needed.

---

> > > > > > ### Comment · Reviewer_PpXk · 2025-08-08
> > > > > > **Thank you for additional experiments**
> > > > > >
> > > > > > Thank you very much for providing the additional experiments.
> > > > > > My concerns have been fully addressed. I will raise my score.

---

### Official Review · Reviewer_K1n9 · 2025-07-03

**Clarity:** 3
**Significance:** 2
**Originality:** 2
**Rating:** 4
**Confidence:** 3

**Summary:**

This paper introduces Adversary Aware Optimization (AAOpt), a test-time defense framework that combines a pretrained diffusion prior with a learned perturbation score model to perform input purification. The method formulates the purification process as a MAP estimation problem, integrating both the data prior and an adversarial likelihood term. The likelihood is modeled by a lightweight score network trained on adversarial residuals. Compared with prior diffusion-based defenses, the proposed method demonstrates improved robust accuracy under strong adaptive attacks on CIFAR-10 and CIFAR-100.

**Questions:**

See the weaknesses given above.

**Ethical Concerns:**

["NO or VERY MINOR ethics concerns only"]

**Final Justification:**

The authors have addressed some concerns, but not all. I will maintain my original rating, as it already accounted for some possibly positive responses from the authors.

**Limitations:**

The paper briefly mentions test-time cost and one-time adversarial data generation as limitations, but it does not discuss the lack of evaluations on more complex datasets or the theoretical gap in the objective derivation. Including a clearer discussion of generalization and theoretical limitations would help strengthen the paper.

**Quality:**

3

**Strengths And Weaknesses:**

Strengths:

The paper addresses a critical issue in adversarial purification, namely the failure of diffusion models to account for structured adversarial perturbations, by explicitly incorporating an adversarial score model.  The proposed optimization framework is efficient and avoids full reverse diffusion sampling.  Experimental results show consistent improvement over both adversarial training and prior purification methods across CIFAR-10 and CIFAR-100, under PGD-EOT and BPDA-EOT attacks.

Weaknesses:
1. **Limited Dataset Scope**
The experiments are restricted to CIFAR-10 and CIFAR-100. Previous works on purification (e.g., DiffPure, ScoreOpt) often include TinyImageNet and ImageNet subsets to better evaluate scalability and generalization. The lack of larger-scale datasets raises doubts about the method’s applicability beyond small resolution benchmarks.

2. **Unclear Derivation in Method Section**
The derivation of Equation (4) appears problematic. Specifically, the term involving `−ϵ` in the surrogate for
\[
\log p(x) \propto x^\top \left[ s_\theta(x_t, t) - \epsilon \right]
\]
is not rigorously justified and seems to be heuristically adapted from Tweedie’s formula. Given that this surrogate plays a central role in the optimization objective, a more precise derivation or theoretical justification is needed.

3. **One-Time Attack Generation Overhead**
The proposed perturbation score model, although efficient, requires a full round of adversarial data generation prior to test-time deployment. This pre-processing cost is non-trivial and could limit scalability.

4. **Hyperparameter Sensitivity**
Although some ablation studies are presented, it remains unclear how sensitive the method is to the choice of \( \lambda \) and other hyperparameters, especially on unseen architectures.

5. **Lack of Comparison with Stronger Adaptive Attacks**
Although PGD-EOT and BPDA-EOT are included, the paper does not evaluate against more recent and stronger purification-aware adaptive attacks such as done in [1].

6. **Unclear Scalability to High-Resolution Tasks**
The method’s practicality on high-resolution datasets remains questionable. The reliance on pixel-level gradient updates and score models trained at small resolution may not transfer well to more realistic image sizes used in safety-critical applications.

7. **Limited Theoretical Depth**
While the method is motivated from a MAP perspective, the paper lacks a rigorous theoretical analysis regarding convergence guarantees, stability of the optimization procedure, or robustness bounds. This makes it difficult to assess the theoretical soundness of the proposed surrogate objective.

[1] DiffAttack: Evasion Attacks Against Diffusion-Based Adversarial Purification, NeurIPS 2024

---

> ### Author Rebuttal · Authors · 2025-07-31
>
> Dear Reviewer,
>
> Thank you for your thoughtful review. We address your concerns below.
>
> We submitted our appendix as a zip file with the supplementary materials, following one of the two allowed methods for NeurIPS submissions. Although we understand you are not obligated to review it, some content couldn’t fit in the main paper, necessitating its inclusion there. The appendix answers most of your questions, and we hope you look at it.
>
> *1. "Limited Dataset Scope "*
>
> We included a comparison with Score-Opt under the same setup on the TinyImageNet dataset (Section 6.1.5). While the results could potentially be improved further by using stronger diffusion models and classifiers, our method already outperforms Score-Opt under the exact same conditions.
>
> *2. "Unclear Derivation in Method Section "*
>
> We will derive the equation below. There was a minor error when writing the equation, which we will fix below and in the camera-ready version if accepted.
>
> Using Taylor expansion of $p_t$ distribution around $\mathbf{x}_t$:
>
> $\log p_t(\mathbf x) \approx \log p_t(\mathbf x_t) + (\mathbf x-\mathbf x_t)^T \nabla_{\mathbf x_t} \log p_t(\mathbf x_t) = \log p_t(\mathbf x_t) - \sigma_t \boldsymbol{\epsilon}^T s_\theta(\mathbf x_t, t)$
>
> Using Bayes rule, we have:
>
> $\log p(\mathbf x) \approx \log p_t(\mathbf x_t) + \log p(\mathbf x| \mathbf x_t) + \frac{1}{2\sigma_t^2}|\mathbf x- \mathbf x_t|^2$
>
> By replacing $\log p_t(\mathbf x_t)$ from the Taylor approximation, we get:
>
> $\log p(\mathbf x) \approx \sigma_t \boldsymbol{\epsilon}^T s_\theta(\mathbf x_t, t) + \frac{1}{2\sigma_t^2}|\mathbf x- \mathbf x_t|^2 + \log p_t(\mathbf x) + \log p(\mathbf x|\mathbf x_t)$
>
> The MAP estimate of $\log p(x)$ would have the same solution as:
>
> $\log p(\mathbf x) \propto \sigma_t \boldsymbol{\epsilon}^T s_\theta(\mathbf x_t, t) + \frac{1}{2\sigma_t^2}|\mathbf x- \mathbf x_t|^2$
>
> which can be rephrased as:
>
> $\log p(\mathbf x) \propto \sigma_t \epsilon^T s_\theta(\mathbf x_t, t) + \frac{1}{2} \boldsymbol \epsilon^T \boldsymbol \epsilon = \epsilon^T(\sigma_t s_\theta(\mathbf x_t, t) + \frac{1}{2}\epsilon) = - \frac{1}{\sigma_t}(x - x_t)^T(\sigma_t s_\theta(\mathbf x_t, t) + \frac{1}{2}\epsilon) = \mathbf x^T (-s_\theta(\mathbf{x}t,t) - \frac{1}{\sigma_t} \boldsymbol \epsilon) + \mathbf x_t^T (s\theta(\mathbf{x}_t,t) + \frac{1}{\sigma_t} \boldsymbol \epsilon)$
>
> Since we only care about the MAP estimate on $p(\mathbf x)$, we can ignore the last term as it does not directly depend on $\mathbf x$.
>
> Therefore:
>
> $\log p(\mathbf x) \propto \mathbf x^T (-\sigma_t s_\theta(\mathbf{x}_t,t) - \boldsymbol \epsilon)$
>
>
> *3. "One-Time Attack Generation Overhead "*
>
> We do not require adversarial generation for test-time, but only during training of our perturbation model. And
> compared to adversarial training papers that continuously generate adversarial examples, our method only requires the initial one-time generation of adversarial examples. Even though this may have some cost, it's limited and can be saved once and reused again and again during training, which significantly reduces the cost. And this is not required for test-time, it's only required once during training of the perturbation model. Crucially, no adversarial generation is needed during inference—only optimization is performed at test time.
>
> *4. "Hyperparameter Sensitivity "*
>
> As shown in our ablation study (Figure 2), performance improves consistently over the prior diffusion as the hyperparameter $\lambda$ varies within the range 0 to 1. Similar to other hyperparameters in deep learning, $\lambda$ can be selected using a validation set (or a subset of the training data if no separate validation set is available). We observe that the optimization is generally stable, and tuning $\lambda$ is not particularly sensitive or difficult in practice.
>
> *5. Lack of Comparison with Stronger Adaptive Attacks
> Although PGD-EOT and BPDA-EOT are included, the paper does not evaluate ...*
>
> We used PGD-EOT because it is currently one of the strongest attacks. In [1], it can be seen that the Robust Accuracy of DiffPure under PGD-EOT is lower than the Robust Accuracy of DiffPure under DiffAttack, showing that PGD-EOT is stronger than DiffAttack, which is proposed by [2]. We also observed that our model performs well under DiffAttack, which we have shown below. We used the same setup and hyperparameters as our model used for the PGD-EOT attack.
>
> --------------------------------------------------
>  |       AAOpt Clean Acc |    91.6                  |
>
>  |    AAOpt Robust Acc  |    87.5                   |
>
> Tab1: DiffAttack AAOpt Performance on $l_\infty(\epsilon = \frac{8}{255})$ using WRN-28-10
>
> *6. "Unclear Scalability to High-Resolution Tasks "*
>
> We have demonstrated our results and comparisons on widely used adversarial benchmarks such as CIFAR-10 and CIFAR-100, and additionally included TinyImageNet to extend the evaluation. As long as a diffusion model is well-trained on higher-resolution datasets, our framework remains applicable. Naturally, scaling any generative model to high-resolution tasks presents challenges, and generating adversarial examples at those scales—especially with iterative gradient-based methods—can be computationally intensive. This is also why many prior works limit evaluations to smaller datasets. Nonetheless, the focus of our current study is to address adversarial robustness on standard, widely-used benchmarks.
>
> *7. "Limited Theoretical Depth"*
>
> While our method is motivated by a MAP formulation, we acknowledge that the paper does not present formal convergence guarantees or robustness bounds. However, our approach is rooted in established likelihood maximization principles commonly used in generative modeling and optimization literature. We have expanded the theoretical background in the revised version to clarify the underlying rationale and assumptions. In practice, our optimization procedure is stable, as supported by extensive empirical results.
>
> Although we do not provide formal robustness bounds, we believe this does not undermine the soundness of the approach. Moreover, the response to Question 2 provides further insight into the surrogate objective and its practical generalization.
>
> We hope the above responses have clarified your concerns. Many of the points raised were also addressed in the appendix, and we appreciate your comments. Let us know if you have any further questions, and we kindly ask you to reconsider your score in light of these clarifications.
>
> 1. Lee, M., & Kim, D. (2023). Robust evaluation of diffusion-based adversarial purification. In Proceedings of the IEEE/CVF International Conference on Computer Vision (pp. 134-144).
>
> 2. Kang, M., Song, D., & Li, B. (2023). Diffattack: Evasion attacks against diffusion-based adversarial purification. Advances in Neural Information Processing Systems, 36, 73919-73942.

---

### Official Review · Reviewer_Ds3y · 2025-07-03

**Clarity:** 2
**Significance:** 3
**Originality:** 4
**Rating:** 5
**Confidence:** 4

**Summary:**

The paper proposes a new adversarial defense method through adversarial purification that recovers the clean image from a perturbed observation. It uses two diffusion models to help construct the objective function for optimizing the clean image estimation. The proposed method is compared against a series of adversarial defense methods, including both adversarial training and adversarial purification, through benchmark testing. The results demonstrate superior robust accuracy with satisfactory standard accuracy for the proposed method, under different attacks.

**Questions:**

I invite the authors to clarify a couple of things as below. The answers will help me assess the quality, particularly on “Is the submission technically sound?”, based on which I will adjust my score.

Can the authors expand on how to derive Eq. (4) by substituting the two mentioned equations?

According to Section 2.3, the two diffusion models for modelling clean data and perturbation distributions are both score based, i.e., both s_{\theta} and d_{\phi} predict the score functions. However, according to Section 4, the used pretrained diffusion model for clean data predicts the denoised estimate. Can the authors explain how do they adapt such a predictor for score function prediction? Also, according to Algorithm 2 in Appendix, the adversarial predictor d_{\phi} predicts the noise (line 7). Can the authors explain why it is ok to use this noise predictor as the score model required by Eq. (5)?

In line 174/175, it mentions that both clear and adversarial examples can be selected to pair with a clean image. But line 2 of Algorithm 2 in Appendix does not seem to reflect this. Can the authors check and explain this?

When preparing image pairs for training the perturbation diffusion model, by randomly selecting images to pair, there is a chance to produce image pairs each containing two very different images. Isn’t this violating the perturbation strength constraint, e.g., roughly ||x - \hat{x}|| <= \epsilon? Can the authors comment on this?

**Ethical Concerns:**

["NO or VERY MINOR ethics concerns only"]

**Final Justification:**

The authors clarified my questions, and responded on experiment side. Based on these I raise my rating.

**Limitations:**

Yes, there is limitation discussion in appendix.

**Quality:**

3

**Strengths And Weaknesses:**

Strengths:

The paper has introduced a novel adversarial purification method, advancing the research area of adversarial defence. The proposed method is elegant, achieving good performance. The paper is in general well-written, easy to follow.

Weaknesses:

There are a few technical details that are not exactly clear and  require explanation and clarification. For instance, some more details on stop-gradient and how the objective function is optimised would be helpful. Also see my question later.

All evaluations and experiment design are aimed to demonstrate the success of the proposal. There is no experiment dedicated to evaluate/analyse in depth the weakness or failure cases of the proposed method.

---

> ### Author Rebuttal · Authors · 2025-07-31
>
> Dear Reviewer,
>
> Thank you for your thoughtful review. We address your concerns below.
>
> *1. "Can the authors expand on how to derive Eq. (4) by ..."*
>
> We will derive the equation below. There was a minor error when writing the equation, which we will fix below and in the camera-ready version if accepted.
>
> Using Taylor expansion of $p_t$ distribution around $\mathbf{x}_t$:
>
> $\log p_t(\mathbf x) \approx \log p_t(\mathbf x_t) + (\mathbf x-\mathbf x_t)^T \nabla_{\mathbf x_t} \log p_t(\mathbf x_t) = \log p_t(\mathbf x_t) - \sigma_t \boldsymbol{\epsilon}^T s_\theta(\mathbf x_t, t)$
>
> Using Bayes rule, we have:
>
> $\log p(\mathbf x) \approx \log p_t(\mathbf x_t) + \log p(\mathbf x| \mathbf x_t) + \frac{1}{2\sigma_t^2}|\mathbf x- \mathbf x_t|^2$
>
> By replacing $\log p_t(\mathbf x_t)$ from the Taylor approximation, we get:
>
> $\log p(\mathbf x) \approx \sigma_t \boldsymbol{\epsilon}^T s_\theta(\mathbf x_t, t) + \frac{1}{2\sigma_t^2}|\mathbf x- \mathbf x_t|^2 + \log p_t(\mathbf x) + \log p(\mathbf x|\mathbf x_t)$
>
> The MAP estimate of $\log p(x)$ would have the same solution as:
>
> $\log p(\mathbf x) \propto \sigma_t \boldsymbol{\epsilon}^T s_\theta(\mathbf x_t, t) + \frac{1}{2\sigma_t^2}|\mathbf x- \mathbf x_t|^2$
>
> which can be rephrased as:
>
> $\log p(\mathbf x) \propto \sigma_t \epsilon^T s_\theta(\mathbf x_t, t) + \frac{1}{2} \boldsymbol \epsilon^T \boldsymbol \epsilon = \epsilon^T(\sigma_t s_\theta(\mathbf x_t, t) + \frac{1}{2}\epsilon) = - \frac{1}{\sigma_t}(x - x_t)^T(\sigma_t s_\theta(\mathbf x_t, t) + \frac{1}{2}\epsilon) = \mathbf x^T (-s_\theta(\mathbf{x}t,t) - \frac{1}{\sigma_t} \boldsymbol \epsilon) + \mathbf x_t^T (s\theta(\mathbf{x}_t,t) + \frac{1}{\sigma_t} \boldsymbol \epsilon)$
>
> Since we only care about the MAP estimate on $p(\mathbf x)$, we can ignore the last term as it does not directly depend on $\mathbf x$.
>
> Therefore:
>
> $\log p(\mathbf x) \propto \mathbf x^T (-\sigma_t s_\theta(\mathbf{x}_t,t) - \boldsymbol \epsilon)$
>
>
> *2. "According to Section 2.3, the two diffusion model ... "*
>
> Thank you for the observation. Since we can extract the noise $\epsilon_\theta$ from the score function $s_\theta$ regardless of how the diffusion/score/denoiser model was trained, we used the terms somewhat loosely in the main text. To clarify: we indeed extract $\epsilon_\theta$ for both the prior and perturbation terms in our framework, finally.
>
> Score-based models and diffusion models are part of the same family of generative models that transform noise into a data distribution [3,4]. The score $\nabla_{x_t} \log p_t(x_t)$ can be expressed in terms of the noise as $\nabla_{x_t} \log p_t(x_t) = -\frac{1}{\sigma(t)}\epsilon $, which can be easily derived by differentiating the Gaussian distribution $\log p_t(x_t)$ with respect to $x_t$. We extract the $\epsilon_{\theta}$ from the prior diffusion model which predicts the denoised estimate as $\epsilon_{\theta} = \frac{x_t - D(x_t)}{\sigma(t)}$ where $D(x_t)$ is the denoised estimate [5]. We trained a model that directly predicts the $\epsilon_{\theta}$ for the perturbation model, so we can directly take $\epsilon_{\theta}$ for that part.
>
> *3. "In line 174/175, it mentions that both ... "*
>
> Thank you for reviewing our paper in detail. In fact, you are the only person who read the appendix well, and we thank you for that. Algorithm 2 at the top says ${x^k}_{k=1}^K$ is composed by combining clean and adversarial sets. So, even if line 2 labels it as such, the set is composed of the combination. We do this because in the real-world application of our method, we don't know if a given $\hat{x}$ is clean or adversarial. To ensure that the perturbation model also learns about data that doesn't have any adversarial perturbation, we included the clean examples in the set.
>
>
> *4. "When preparing image pairs for training ..."*
>
> The image pairs are created by randomly selecting samples from the dataset, but they are originally constructed as pairs — each image has both a clean version and an adversarial (or possibly clean) counterpart. This ensures that each pair corresponds to the same base image, maintaining consistency during training.
>
> We hope the above answers clarified your questions. We will be happy to answer if you have any further questions.
>
>
> 1. Mardani, M., Song, J., Kautz, J., & Vahdat, A. (2023). A variational perspective on solving inverse problems with diffusion models. ICLR 2024. \\
> 2. Vahdat, A., Kreis, K., & Kautz, J. (2021). Score-based generative modeling in latent space. Advances in neural information processing systems, 34, 11287-11302. \\
> 3. Song, Y., Sohl-Dickstein, J., Kingma, D. P., Kumar, A., Ermon, S., & Poole, B. (2020). Score-based generative modeling through stochastic differential equations. ICLR 2021. \\
> 4. Ho, J., Jain, A., & Abbeel, P. (2020). Denoising diffusion probabilistic models. Advances in neural information processing systems, 33, 6840-6851. \\
> 5. Karras, T., Aittala, M., Aila, T., & Laine, S. (2022). Elucidating the design space of diffusion-based generative models. Advances in neural information processing systems, 35, 26565-26577.

---

> > ### Comment · Reviewer_Ds3y · 2025-08-04
> >
> > I thank the authors for the detailed response to my questions, which are highly appreciated. About Eq. (4), since the derived form in the response is slightly different, Eq. (5) and Algorithm 1 seem to need some light modifications. Doesn't this affect the implementation and the results?

---

> ### Author Response · Authors · 2025-08-04
> **Reply to Comment**
>
> Dear Reviewer,
>
> Thank you for the follow-up. As you have pointed out, there are some slight differences, but those occurred due to last-minute editing and a mixing up of usage of the score $s_\theta$ and the $\epsilon_{\theta}$ diffusion models. They have no effect on the experiments, as all the experiments followed the correct version, which is also available in other works, specifically in [1]. Around lines 160 and 161, we have listed that equation 4 should be similar to the objectives used by [1]. And if you see [1], they have a similar setup, but they have only used it for their prior diffusion for an inverse task, in contrast to our work, which is set up for adversarial defense with a perturbation model. Our experiments have the same format as theirs, which can also be verified in our attached code. Compared to their work, we have a different setup with a perturbation model applied to the task of adversarial defense. That's why we offered to fix the equations and add the derivation in the appendix in the final version without affecting any of the results of the experiments. And [1] also derived their equations from a variational perspective different from ours, which can be seen as a different way of deriving the equations. We appreciate your follow-up immensely.
>
> [1] Mardani, M., Song, J., Kautz, J., & Vahdat, A. A variational perspective on solving inverse problems with diffusion models. ICLR 2024.

---

> > ### Comment · Reviewer_Ds3y · 2025-08-05
> >
> > Thank you for the reply. Given the confirmation and clarification on things, I keep holding my positive view on the paper, and will not reduce my rating. I hesitate to raise my rating to 5, as there is space to improve on experiment side in the submitted version.

---

> > > ### Comment · Reviewer_Ds3y · 2025-08-05
> > >
> > > I overlooked some response to experiments to other reviewers.  Considering those, I raise my rating to 5.

---

> > > > ### Author Response · Authors · 2025-08-05
> > > > **Acknowledgement**
> > > >
> > > > Dear Reviewer, we appreciate your review. Thank you for your feedback.

---

### Note · Authors · 2025-08-11

Dear Reviewers and Chairs,

We sincerely thank you for the constructive and engaging discussion during the rebuttal phase. As presented, our proposed method (AAOpt) introduces a principled framework for optimizing adversarial examples by jointly leveraging a prior and a perturbation model. This approach achieves notable robustness gains over existing methods, rigorously evaluated against strong adaptive attacks.

Although we addressed all the points raised during the discussion, we summarize the key clarifications here for completeness:

- Equation corrections (Reviewer Ds3y, also raised by Reviewer K1n9): As noted in our rebuttal, if accepted, we will correct minor equation mistakes in the camera-ready version. These do not affect any reported experimental results, as the equations in question are part of the surrogate derivation. The correct formulations—used in all experiments—are consistent with prior work [1] and are already cited in our paper and rebuttals.

- Training and inference details (Reviewer Ds3y, Reviewer K1n9): We have addressed the details during the discussion period on how we train the perturbation model and our inference method.


- Additional experiments: For questions not already addressed in the appendix (Reviewers K1n9, PpXk, and 9umZ), we provided additional results in the rebuttal, including:

     - Adding stronger attacks such as DiffAttack [2]

     - Adding the ADBM baseline for direct comparison [3]

     - Ablation studies on norm-transfer performance

- Certified defense questions (Reviewers K1n9 and PpXk): Clarified that our primary contribution is a robust optimization objective grounded in theoretical principles, with certified robustness considered as an important avenue for future work.

We believe these clarifications, together with the strong empirical evidence obtained from rigorous evaluations against strong adaptive attacks, further reinforce the contribution and readiness of our work.

[1] Mardani, M., Song, J., Kautz, J., & Vahdat, A. A variational perspective on solving inverse problems with diffusion models. ICLR 2024.

[2] Kang, M., Song, D., & Li, B. (2023). Diffattack: Evasion attacks against diffusion-based adversarial purification. Advances in Neural Information Processing Systems, 36, 73919-73942.

[3] Li, X., Sun, W., Chen, H., Li, Q., Liu, Y., He, Y., ... & Hu, X. (2024). Adbm: Adversarial diffusion bridge model for reliable adversarial purification. ICLR 2025.

---

### Decision · Program_Chairs · 2025-09-17

**Decision:**

Accept (poster)

**Comment:**

This paper proposes Adversary Aware Optimization (AAOpt), an adversarial purification framework for defending deep neural networks against adversarial attacks. The key innovation lies in explicitly modeling adversarial perturbations using a lightweight score-based diffusion model, which is combined with a pretrained diffusion prior to guide the purification process. The method formulates purification as a Maximum A Posteriori (MAP) estimation problem, integrating both the clean data distribution and adversarial perturbation structure. Experiments on CIFAR-10/100 demonstrate improved robust accuracy against adaptive attacks compared to existing adversarial training and purification baselines. The authors also validate generalization across classifier architectures and perturbation norms.

This paper has several strengths. It shows strong defense against adaptive attacks, with ablation studies showing consistent improvements over baselines, such as DiffPure and Score-Opt. The modular design allows the perturbation model to be adjusted, offering adaptability to different threat scenarios. The generalization is evidenced by robustness across classifier architectures and perturbation norms. However, the reviewers also raised several concerns. The scope of evaluation is limited, where the evaluation is conducted on small-resolution datasets, and the scalability to high-resolution tasks remains unverified. Additionally, the derivation of the surrogate objective (Eq. 4) lacks rigorous justification, and no formal robustness guarantees are provided. Also, the computational overhead has not been clearly stated. While efficient compared to full reverse diffusion, the inference-time cost is not thoroughly benchmarked.

During the rebuttal phase, the authors have addressed most concerns raised by the reviewers. Specifically, Reviewer Ds3y and K1n9 questioned the derivation of Eq. 4 and score/noise predictor consistency. The authors provided detailed derivations and clarified implementation alignment with prior work. Reviewer PpXk raised concern about the incremental contribution and missing comparisons to ADBM. The authors added experiments showing AAOpt’s superiority over ADBM. Reviewer 9umZ questioned about cross-norm and cross-architecture transferability. The authors provided new results $l_\infty$-trained model tested on $l_2$ attacks. Although some concerns remain after the rebuttal, such as evaluation on high-resolution tasks, most concerns have been addressed, and all the reviewers lean towards acceptance.

The paper’s empirical contributions and further clarifications outweigh its limitations, making it a solid addition to the adversarial robustness literature. The authors are encouraged to improve the paper by considering the reviewers’ comments carefully.